# Named Tensor Notation

**David Chiang**
*University of Notre Dame*

**Alexander M. Rush**
*Cornell University*

**Boaz Barak**
*Harvard University*

**Reviewed on OpenReview:** *https://openreview.net/forum?id=hVT7SHlilx*

## Abstract

We propose a notation for tensors with named axes, which relieves the author, reader, and future implementers of machine learning models from the burden of keeping track of the order of axes and the purpose of each. The notation makes it easy to lift operations on low-order tensors to higher order ones, for example, from images to minibatches of images, or from an attention mechanism to multiple attention heads.

After a brief overview and formal definition of the notation, we illustrate it through several examples from modern machine learning, from building blocks like attention and convolution to full models like Transformers and LeNet. We then discuss differential calculus in our notation and compare with some alternative notations. Our proposals build on ideas from many previous papers and software libraries. We hope that our notation will encourage more authors to use named tensors, resulting in clearer papers and more precise implementations.

## 1  Introduction

Formal descriptions of neural networks primarily adopt the notation of vectors and matrices from applied linear algebra (Goodfellow et al., 2016). When used to describe vector spaces, this notation is both concise and unambiguous. However, when applied to neural networks, these properties are lost. Consider the equation for attention as notated in the Transformer paper (Vaswani et al., 2017):

$$\text{Attention}(Q, K, V) = \left( \text{softmax} \frac{QK^\top}{\sqrt{d_k}} \right) V.$$

The equation relates $Q$, $K$, and $V$ (for query, key, and value, respectively) as sequences of feature vectors, packed into possibly identically-sized matrices. While concise, this equation is ambiguous. Does the product $QK^\top$ sum over the sequence, or over the features? We know that it sums over columns, but there is not enough information to know what the columns represent. Is the softmax taken over the query sequence or the key sequence? The usual notation does not offer an answer. Perniciously, the implementation of an incorrect interpretation might still run without errors. With the addition of more axes, like multiple attention heads or multiple sentences in a minibatch, the notation becomes even more cumbersome.

We propose an alternative mathematical notation for tensors with *named axes*.[1] The notation has a formal underpinning, but is hopefully intuitive enough that machine learning researchers can understand it without much effort. In named tensor notation, the above equation becomes

$$\text{Attention} \colon \mathbb{R}^{\mathsf{key}} \times \mathbb{R}^{\mathsf{seq} \times \mathsf{key}} \times \mathbb{R}^{\mathsf{seq} \times \mathsf{val}} \to \mathbb{R}^{\mathsf{val}}$$

---

[1] We follow NumPy in using the term *axis*. Other possible terms would be *index*, *dimension*, *way*, or *mode* (Tucker, 1964), but we felt that *axis* had the least potential for confusion.

$$\text{Attention}(Q, K, V) = \left(\underset{\text{seq}}{\text{softmax}} \frac{Q \underset{\text{key}}{\odot} K}{\sqrt{|\text{key}|}}\right) \underset{\text{seq}}{\odot} V.$$

The type signature introduces three named axes: the key axis is for features of queries and keys, the val axis is for features of values, and the seq axis is for tokens in a sequence. (Please see Section 2.2 for an explanation of our naming convention.) This notation makes the types of each input tensor explicit. Tensor $Q$ is a query vector that is compared with key vectors, so it has a key axis. Tensor $K$ is a sequence of key vectors, so it has seq and key axes. Tensor $V$ is a sequence of value vectors, so it has seq and val axes. Unlike with matrix notation, the reader is not required to remember whether seq corresponds to rows or columns in either of these tensors.

The function itself uses the named axes to precisely apply operations. The expression $Q \underset{\text{key}}{\odot} K$ is a dot product over the key axis shared between $K$ and $Q$; there is no ambiguity about rows or columns. Similarly, the softmax function is annotated with the axis along which it is applied, removing any ambiguity or reliance on convention.

Furthermore, named tensor notation naturally extends to *lifting* (also known as vectorizing and/or broadcasting) a function to tensors with more axes. For example, if instead of being a tensor with the single axis key, $Q$ has three axes key, seq and batch (corresponding to tokens of a sequence and examples in a minibatch, respectively) then the Attention function works as written, acting on each example in a minibatch in parallel. Similarly, we can also add a heads axis to the inputs to get multiple attention heads. These additional axes are often elided in neural network papers, possibly avoiding notational complexity, but possibly also hiding critical model details.

**Our contributions.** This work proposes a *mathematical notation* for named tensors and a fully specified *semantic interpretation* for the notation. Through examples, we demonstrate that this notation enables specifying machine learning models and operations in a succinct yet precise manner. The need for named tensors has been recognized by several software packages, including xarray (Hoyer & Hamman, 2017), Nexus (Chen, 2017), tsalib (Sinha, 2018), axisarrays (Bauman, 2018), NamedTensor (Rush, 2019), PyTorch (Torch Contributors, 2019), Dex (Paszke et al., 2021), JAX (JAX authors, 2021), einops (Rogozhnikov, 2022), and torchdim (DeVito, 2023). While our notation is inspired by these efforts, our focus is on mathematical notation to be used in papers, whereas previous efforts have focused on code. Our hope is that our notation will be adopted by authors, leading to clearer, more replicable papers, and that this, in turn, will encourage more implementers to adopt named tensor libraries, leading to clearer, more correct code.

## 2 Named Tensors

In standard notation, a vector, matrix, or tensor is indexed by an integer or sequence of integers; if it has dimensions $n_1, \ldots, n_r$, it can be thought of as a map that takes as input $(i_1, \ldots, i_r) \in [n_1] \times \cdots \times [n_r]$ and outputs a real number (or an element of a different field). For example, if $A \in \mathbb{R}^{3 \times 3}$, then the order of the two axes matters: $A_{1,3}$ and $A_{3,1}$ are not the same element. It is up to the reader to remember what each axis of each tensor stands for. This problem is exacerbated in modern machine learning, where tensors have multiple axes with different meanings (batches, channels, etc.), and different operations act on different axes.

In contrast, we propose *named tensors*, in which each axis has a *name* that describes it and ensures there is no confusion between axes. We write $\mathsf{ax}[n]$ for an axis with name $\mathsf{ax}$ and size $n$, and we write $\mathsf{ax}(i)$ to index the $i$-th element along axis $\mathsf{ax}$. So if a tensor has axes $\mathsf{ax}_1[n_1], \ldots, \mathsf{ax}_r[n_r]$ (with $\mathsf{ax}_1, \ldots, \mathsf{ax}_r$ being distinct names), it can be thought of as a map that takes as input a *record* $\{\mathsf{ax}_1(i_1), \ldots, \mathsf{ax}_r(i_r)\}$, with $i_1 \in [n_1], \ldots, i_r \in [n_r]$, and outputs a field element.

In summary the key difference is that, while a tensor in standard notation takes as input an ordered tuple of indices, a named tensor takes as input a record, which is an unordered set of named indices. We illustrate with some examples below, then give formal definitions.

## 2.1 By example

For example, if $A$ represents a $3 \times 3$ grayscale image, we can make it a named tensor like so (writing it two equivalent ways to show that the order of axes does not matter):

$$A \in \mathbb{R}^{\mathsf{height}[3] \times \mathsf{width}[3]} = \mathbb{R}^{\mathsf{width}[3] \times \mathsf{height}[3]}$$

$$A = \begin{matrix} & \mathsf{width} \\ \mathsf{height} & \begin{bmatrix} 3 & 1 & 4 \\ 1 & 5 & 9 \\ 2 & 6 & 5 \end{bmatrix} \end{matrix} = \begin{matrix} & \mathsf{height} \\ \mathsf{width} & \begin{bmatrix} 3 & 1 & 2 \\ 1 & 5 & 6 \\ 4 & 9 & 5 \end{bmatrix} \end{matrix}. \tag{1}$$

We access elements of $A$ using named indices, whose order again does not matter: $A_{\mathsf{height}(1),\mathsf{width}(3)} = A_{\mathsf{width}(3),\mathsf{height}(1)} = 4$. We also allow partial indexing:

$$A_{\mathsf{height}(1)} = \begin{matrix} \mathsf{width} \\ \begin{bmatrix} 3 & 1 & 4 \end{bmatrix} \end{matrix} \qquad\qquad A_{\mathsf{width}(3)} = \begin{matrix} \mathsf{height} \\ \begin{bmatrix} 4 & 9 & 5 \end{bmatrix} \end{matrix}.$$

It does not matter if we write $A_{\mathsf{height}(1)}$ or $A_{\mathsf{width}(3)}$ as row and column vectors. In many contexts, an axis name is used with only one size. If so, we can simply write $\mathsf{height}$ for the unique axis with name $\mathsf{height}$, as in $\mathbb{R}^{\mathsf{height} \times \mathsf{width}}$. We can leave the size of an axis unspecified at first, and specify its size later (e.g., deferring it to an appendix on experimental details). For example, we can specify $|\mathsf{height}| = |\mathsf{width}| = 28$ if we want to prescribe the precise size of an image, or just write $|\mathsf{height}| = |\mathsf{width}|$ to specify that it's a square image.

## 2.2 What's in a name?

Although users of this notation are free to choose any names for axes, we offer the following recommendations. First, we recommend *words* instead of single letters, to communicate better the meaning of each axis.

More subtly, we recommend words that describe a *whole* rather than its parts. For example, to represent a minibatch of examples, we would name the axis $\mathsf{batch}$; to represent a sequence of tokens, we would name the axis $\mathsf{seq}$. One reason for this choice is that there are cases, like $\mathsf{height}$ and $\mathsf{width}$, where there is a name for the whole, but no unambiguous name for the part. By contrast, in cases where there is a name for the part but not the whole, it's always possible to use the plural form of the name of the part. For example, if we wanted $A$ to have red, green, and blue channels, we would name the axis $\mathsf{chans}$.

Section 4 contains many more examples of axis names.

## 2.3 Formal definition

We now define formally the notation we use.

**Definition 1** (Names, indices, and axes)**.** An *axis* is a pair, written $\mathsf{ax}[I]$, where

- $\mathsf{ax}$ is the *name* of the axis, which is simply a string of letters. We write both names and variables ranging over names using sans-serif font.

- $I$ is a set of *indices*. In this paper, $I$ is always of the form $\{1, \ldots, n\}$ for some $n$, so we abbreviate $\mathsf{ax}[\{1, \ldots, n\}]$ as $\mathsf{ax}[n]$.

In many contexts, there is only one axis with name $\mathsf{ax}$, and so we refer to the axis simply as $\mathsf{ax}$. The context always makes it clear whether $\mathsf{ax}$ is a name or an axis. If $\mathsf{ax}$ is an axis, we write $\mathrm{ind}(\mathsf{ax})$ for its index set, and we write $|\mathsf{ax}|$ as shorthand for $|\mathrm{ind}(\mathsf{ax})|$.

**Definition 2** (Named indices and records)**.** If $\mathsf{ax}[I]$ is an axis and $i \in I$, then a *named index* is a pair, written $\mathsf{ax}(i)$. A *record* is a set of named indices $\{\mathsf{ax}_1(i_1), \ldots, \mathsf{ax}_r(i_r)\}$, where $\mathsf{ax}_1, \ldots \mathsf{ax}_r$ are pairwise distinct names.

**Definition 3** (Shapes). A *shape* is a set of axes, written $\mathsf{ax}_1[I_1] \times \cdots \times \mathsf{ax}_r[I_r]$, where $\mathsf{ax}_1, \ldots \mathsf{ax}_r$ are pairwise distinct names. We write $\emptyset$ for the empty shape. A shape defines a set of records:

$$\mathrm{rec}(\mathsf{ax}_1[I_1] \times \cdots \times \mathsf{ax}_r[I_r]) = \{\{\mathsf{ax}_1(i_1), \ldots, \mathsf{ax}_r(i_r)\} \mid i_1 \in I_1, \ldots, i_r \in I_r\}.$$

We say two shapes $\mathcal{S}$ and $\mathcal{T}$ are *compatible* if whenever $\mathsf{ax}[I] \in \mathcal{S}$ and $\mathsf{ax}[J] \in \mathcal{T}$, then $I = J$. We say that $\mathcal{S}$ and $\mathcal{T}$ are *orthogonal* if there is no $\mathsf{ax}$ such that $\mathsf{ax}[I] \in \mathcal{S}$ and $\mathsf{ax}[J] \in \mathcal{T}$ for any $I$, $J$. If $t \in \mathrm{rec}\,\mathcal{T}$ and $\mathcal{S} \subseteq \mathcal{T}$, then we write $t|_{\mathcal{S}}$ for the unique record in $\mathrm{rec}\,\mathcal{S}$ such that $t|_{\mathcal{S}} \subseteq t$.

**Definition 4** (Named tensors). Let $F$ be a field and let $\mathcal{S}$ be a shape. Then a *named tensor over $F$ with shape $\mathcal{S}$* is a mapping from $\mathrm{rec}\,\mathcal{S}$ to $F$. If $X$ has shape $\mathcal{S}$ then we write $\mathrm{shp}\,X = \mathcal{S}$. We write the set of all named tensors with shape $\mathcal{S}$ as $F^{\mathcal{S}}$.

We don't make any distinction between a scalar (an element of $F$) and a named tensor with empty shape (an element of $F^{\emptyset}$).

If $X \in F^{\mathcal{S}}$, then we access an element of $X$ by applying it to a record $s \in \mathrm{rec}\,\mathcal{S}$; but we write this using the usual subscript notation: $X_s$ rather than $X(s)$. To avoid clutter, in place of $X_{\{\mathsf{ax}_1(i_1), \ldots, \mathsf{ax}_r(i_r)\}}$, we usually write $X_{\mathsf{ax}_1(i_1), \ldots, \mathsf{ax}_r(x_r)}$. When a named tensor is an expression like $(X + Y)$, we index it by surrounding it with square brackets like this: $[X + Y]_{\mathsf{ax}_1(i_1), \ldots, \mathsf{ax}_r(x_r)}$.

We also allow partial indexing. If $X$ is a tensor with shape $\mathcal{T}$ and $s \in \mathrm{rec}\,\mathcal{S}$ where $\mathcal{S} \subseteq \mathcal{T}$, then we define $X_s$ to be the named tensor with shape $\mathcal{T} \setminus \mathcal{S}$ such that, for any $t \in \mathrm{rec}(\mathcal{T} \setminus \mathcal{S})$,

$$[X_s]_t = X_{s \cup t}.$$

(For the edge case $\mathcal{T} = \emptyset$, our definitions for indexing and partial indexing coincide: one gives a scalar and the other gives a tensor with empty shape, but we don't distinguish between the two.)

## 3 Operations

A significant benefit of named tensor notation is that it allows one to unambiguously specify *operations* that map tensors to tensors, and defines precisely how operations can be *lifted* when an operation is applied to tensors with more axes than are present in its signature and how *broadcasting* happens when different arguments add different axes.

We start with the formal definition of named tensor operations and lifting, then show how this definition leads to many common operations.

### 3.1 Formal definition

By *(named) tensor function* or *(named) tensor operation*, we mean not only functions from tensors to tensors, but also operators like negation $(-)$, addition $(+)$, and so on. We will extend the standard function/operator notation by allowing tensor operations to be *lifted* to higher-order tensors.

**Definition 5** (lifting, unary). Let $f \colon F^{\mathcal{S}} \to G^{\mathcal{T}}$ be a function from tensors to tensors. For any shape $\mathcal{S}'$ orthogonal to both $\mathcal{S}$ and $\mathcal{T}$, we can define the *lift* $f^{\mathcal{S}'}$ of $f$ with the shape $\mathcal{S}'$ to be the map

$$f^{\mathcal{S}'} \colon F^{\mathcal{S} \cup \mathcal{S}'} \to G^{\mathcal{T} \cup \mathcal{S}'}$$
$$\left[f^{\mathcal{S}'}(X)\right]_{s'} = f(X_{s'}) \qquad \text{for all } X \in F^{\mathcal{S} \cup \mathcal{S}'} \text{ and } s' \in \mathrm{rec}\,\mathcal{S}'.$$

Usually, we simply write $f$ instead of $f^{\mathcal{S}'}$. That is, for every tensor $X$ with shape $\mathcal{R} \supseteq \mathcal{S}$, we let $f(X) = f^{\mathcal{R} \setminus \mathcal{S}}(X)$.

If $f$ is a multary function, we can lift each of its arguments to larger shapes, and we don't have to add the same axes to all the arguments; an axis present in one argument but not another is *broadcast* from the former to the latter. We consider just the case of two arguments; three or more arguments are analogous.

**Definition 6** (lifting, binary). Let $f\colon F^{\mathcal{S}} \times G^{\mathcal{T}} \to H^{\mathcal{U}}$ be a binary function from tensors to tensors. For any shapes $\mathcal{S}'$ and $\mathcal{T}'$ that are compatible with each other and orthogonal to $\mathcal{S}$ and $\mathcal{T}$, respectively, and such that $\mathcal{S}' \cup \mathcal{T}'$ is orthogonal to $\mathcal{U}$, we can lift $f$ to:

$$f^{\mathcal{S}' \cup \mathcal{T}'}\colon F^{\mathcal{S} \cup \mathcal{S}'} \times G^{\mathcal{T} \cup \mathcal{T}'} \to H^{\mathcal{U} \cup \mathcal{S}' \cup \mathcal{T}'}$$
$$\left[ f^{\mathcal{S}' \cup \mathcal{T}'}(X, Y) \right]_{s'} = f\left( X_{s'|_{\mathcal{S}'}}, Y_{s'|_{\mathcal{T}'}} \right) \qquad \text{for all } X \in F^{\mathcal{S} \cup \mathcal{S}'},\, Y \in F^{\mathcal{T} \cup \mathcal{T}'},\, s' \in \operatorname{rec}(\mathcal{S}' \cup \mathcal{T}').$$

Again, we usually write $f$ instead of $f^{\mathcal{S}' \cup \mathcal{T}'}$.

In the following sections, we present some consequences of the above lifting rules. In particular, we show how they allow one to lift some common operations from operating on scalars, vectors, or matrices to operating on tensors with more axes, and how they correspond to vectorizing and broadcasting (as implemented by NumPy and related packages).

### 3.2 Elementwise operations and broadcasting

Any function from a scalar to a scalar corresponds to a tensor function with signature $F^{\emptyset} \to F^{\emptyset}$. Hence lifting it to any tensor shape, by Definition 5, corresponds to elementwise application. For example, given the logistic sigmoid function,

$$\sigma\colon \mathbb{R} \to \mathbb{R}$$
$$\sigma(x) = \frac{1}{1 + \exp(-x)}$$

we can lift it to tensors. If $A \in \mathbb{R}^{\mathsf{height}[3] \times \mathsf{width}[3]}$ is the example tensor (1), then

$$\sigma(A) = \mathsf{height} \begin{bmatrix} \frac{1}{1+\exp(-3)} & \frac{1}{1+\exp(-1)} & \frac{1}{1+\exp(-4)} \\ \frac{1}{1+\exp(-1)} & \frac{1}{1+\exp(-5)} & \frac{1}{1+\exp(-9)} \\ \frac{1}{1+\exp(-2)} & \frac{1}{1+\exp(-6)} & \frac{1}{1+\exp(-5)} \end{bmatrix} \overset{\mathsf{width}}{}.$$

Similarly for rectified linear units $(\operatorname{relu}(x) = \max(0, x))$, negation, and so on.

Any function with signature $\mathbb{R} \times \mathbb{R} \to \mathbb{R}$, including binary operators like addition $(+)$, can be applied to two named tensors with the same shape. But if we apply a binary function or operator to tensors with different shapes, then, by Definition 6, broadcasting applies. For example, let

$$x \in \mathbb{R}^{\mathsf{height}[3]} \qquad\qquad\qquad y \in \mathbb{R}^{\mathsf{width}[3]}$$

$$x = \mathsf{height} \begin{bmatrix} 2 \\ 7 \\ 1 \end{bmatrix} \qquad\qquad\qquad y = \overset{\mathsf{width}}{\begin{bmatrix} 1 & 4 & 1 \end{bmatrix}}.$$

(We write $x$ as a column just to make the broadcasting easier to visualize.) Then, to evaluate $A + x$, we effectively replace $x$ with a new tensor with a copy of $x$ for every index of axis $\mathsf{width}$. Likewise for $A + y$:

$$A + x = \mathsf{height} \begin{bmatrix} 3+2 & 1+2 & 4+2 \\ 1+7 & 5+7 & 9+7 \\ 2+1 & 6+1 & 5+1 \end{bmatrix} \overset{\mathsf{width}}{} \qquad\qquad A + y = \mathsf{height} \begin{bmatrix} 3+1 & 1+4 & 4+1 \\ 1+1 & 5+4 & 9+1 \\ 2+1 & 6+4 & 5+1 \end{bmatrix} \overset{\mathsf{width}}{}.$$

Similarly for other operations. We write elementwise multiplication (Hadamard product) as $\odot$.

### 3.3 Reductions

The same rules apply to functions from vectors to scalars, called *reductions*. We specify which axis a reduction applies to using a subscript (equivalent to the `axis` argument in NumPy and `dim` in PyTorch), so that even after lifting, we know which axis to reduce. For example, we can define summation:

$$\sum_{\mathsf{ax}} - : \mathbb{R}^{\mathsf{ax}[I]} \to \mathbb{R}$$

$$\sum_{\mathsf{ax}} X = \sum_{i \in I} X_{\mathsf{ax}(i)}$$

and use it on $A$ from Eq. (1):

$$\sum_{\mathsf{height}} A = \sum_{i} A_{\mathsf{height}(i)} = \overset{\mathsf{width}}{\begin{bmatrix} 3+1+2 & 1+5+6 & 4+9+5 \end{bmatrix}}$$

$$\sum_{\mathsf{width}} A = \sum_{j} A_{\mathsf{width}(j)} = \overset{\mathsf{height}}{\begin{bmatrix} 3+1+4 & 1+5+9 & 2+6+5 \end{bmatrix}}.$$

We can also write multiple names to sum over multiple axes:

$$\sum_{\substack{\mathsf{height} \\ \mathsf{width}}} A = \sum_{i} \sum_{j} A_{\mathsf{height}(i),\mathsf{width}(j)} = 3+1+4+1+5+9+2+6+5.$$

But a summation with an index variable (like $i$ or $j$ above) is a standard summation over values of that variable, and a summation with no subscript is a standard summation over a set.

Other examples of reductions include:

$$\operatorname*{norm}_{\mathsf{ax}} X = \sqrt{\sum_{\mathsf{ax}} X^2} \qquad\qquad \operatorname*{norm}_{p}_{\mathsf{ax}} X = \left( \sum_{\mathsf{ax}} X^p \right)^{\frac{1}{p}}$$

$$\min_{\mathsf{ax}} X = \min\{X_{\mathsf{ax}(i)} \mid i \in I\} \qquad\qquad \max_{\mathsf{ax}} X = \max\{X_{\mathsf{ax}(i)} \mid i \in I\}$$

$$\operatorname*{mean}_{\mathsf{ax}} X = \frac{1}{|\mathsf{ax}|} \sum_{\mathsf{ax}} X \qquad\qquad \operatorname*{var}_{\mathsf{ax}} X = \frac{1}{|\mathsf{ax}|} \sum_{\mathsf{ax}} (X - \operatorname*{mean}_{\mathsf{ax}} X)^2$$

### 3.4 Contraction

The vector dot-product is a function from *two* vectors to a scalar. We write it as follows:

$$- \underset{\mathsf{ax}}{\odot} - : \mathbb{R}^{\mathsf{ax}[I]} \times \mathbb{R}^{\mathsf{ax}[I]} \to \mathbb{R}$$

$$X \underset{\mathsf{ax}}{\odot} Y = \sum_{i \in I} X_{\mathsf{ax}(i)} Y_{\mathsf{ax}(i)}$$

When lifted to higher-order tensors, the dot-product generalizes to the ubiquitous *contraction* operator, which can also be thought of as elementwise multiplication followed by summation over one axis, that is,

$$X \underset{\mathsf{ax}}{\odot} Y = \sum_{\mathsf{ax}} X \odot Y. \tag{2}$$

For example,

$$A \underset{\mathsf{height}}{\odot} x = \sum_{\mathsf{height}} A \odot x = \overset{\mathsf{width}}{\begin{bmatrix} 3\cdot 2+1\cdot 7+2\cdot 1 & 1\cdot 2+5\cdot 7+6\cdot 1 & 4\cdot 2+9\cdot 7+5\cdot 1 \end{bmatrix}}$$

$$A \underset{\text{width}}{\odot} y = \sum_{\text{width}} A \odot y = \text{height} \begin{bmatrix} 3 \cdot 1 + 1 \cdot 4 + 4 \cdot 1 \\ 1 \cdot 1 + 5 \cdot 4 + 9 \cdot 1 \\ 2 \cdot 1 + 6 \cdot 4 + 5 \cdot 1 \end{bmatrix}.$$

Again, we can write multiple names to contract multiple axes at once:

$$A \underset{\substack{\text{height} \\ \text{width}}}{\odot} A = \sum_{\substack{\text{height} \\ \text{width}}} A \odot A = 3 \cdot 3 + 1 \cdot 1 + 4 \cdot 4 + 1 \cdot 1 + 5 \cdot 5 + 9 \cdot 9 + 2 \cdot 2 + 6 \cdot 6 + 5 \cdot 5$$

$A \odot$ with no axis name under it contracts zero axes and is equivalent to elementwise multiplication, which is the reason we use the same symbol $\odot$ for elementwise multiplication and contraction. The contraction operator can be used for many multiplication-like operations:

$$x \underset{\text{height}}{\odot} x = \sum_{\text{height}} x \odot x \qquad\qquad \text{inner product}$$

$$x \odot y = \text{height} \overset{\text{width}}{\begin{bmatrix} 2 \cdot 1 & 2 \cdot 4 & 2 \cdot 1 \\ 7 \cdot 1 & 7 \cdot 4 & 7 \cdot 1 \\ 1 \cdot 1 & 1 \cdot 4 & 1 \cdot 1 \end{bmatrix}} \qquad\qquad \text{outer product}$$

$$A \underset{\text{width}}{\odot} y = \sum_{\text{width}} A \odot y \qquad\qquad \text{matrix-vector product}$$

$$x \underset{\text{height}}{\odot} A = \sum_{\text{height}} x \odot A \qquad\qquad \text{vector-matrix product}$$

$$A \underset{\text{width}}{\odot} B = \sum_{\text{width}} A \odot B \qquad\qquad \text{matrix-matrix product } (B \in \mathbb{R}^{\text{width} \times \text{width}'})$$

A contraction of three more tensors can be written as a sum. For example, the three-way inner product of vectors $x, y, z \in \mathbb{R}^{\text{width}}$ can be written as $\sum_{\text{width}} x \odot y \odot z$.

Like the dot-product from which it is lifted, but unlike matrix multiplication, the contraction operator is commutative, but not associative. However, contraction does obey the following associative-like law.

$$X \underset{\mathcal{S} \cup \mathcal{T}}{\odot} (Y \underset{\mathcal{U}}{\odot} Z) = (X \underset{\mathcal{S}}{\odot} Y) \underset{\mathcal{T} \cup \mathcal{U}}{\odot} Z \qquad\qquad \text{if } \mathcal{S} \cap \operatorname{shp} Z = \mathcal{U} \cap \operatorname{shp} X = \emptyset. \qquad (3)$$

The special case

$$X \underset{\mathcal{S}}{\odot} (Y \underset{\mathcal{U}}{\odot} Z) = (X \underset{\mathcal{S}}{\odot} Y) \underset{\mathcal{U}}{\odot} Z \qquad\qquad \text{if } \mathcal{S} \cap \operatorname{shp} Z = \mathcal{U} \cap \operatorname{shp} X = \emptyset \qquad (4)$$

will be useful in Section 5 for moving $Z$ from inside one or more sets of parentheses to the outside.

### 3.5 Vectors to vectors and beyond

Functions from vectors to vectors ($\mathbb{R}^{\text{ax}[I]} \to \mathbb{R}^{\text{ax}[I]}$) lift to functions on tensors that operate along one axis, but leave the tensor shape unchanged. Such functions are particularly problematic in standard notation, which does not provide any way (to our knowledge) of specifying which axis the operation should be performed over. Such functions include:

$$\underset{\text{ax}}{\text{softmax}} X = \frac{\exp X}{\sum_{\text{ax}} \exp X} \qquad\qquad (5a)$$

$$\underset{\text{ax}}{\text{argmax}} X = \lim_{\alpha \to \infty} \underset{\text{ax}}{\text{softmax}} \alpha X \qquad\qquad (5b)$$

$$\underset{\mathsf{ax}}{\operatorname{argmin}} X = \lim_{\alpha \to -\infty} \underset{\mathsf{ax}}{\operatorname{softmax}} \alpha X \tag{5c}$$

For example, we can clearly distinguish between two ways of performing a softmax on $A$:

$$\underset{\mathsf{height}}{\operatorname{softmax}} A = \mathsf{height} \begin{bmatrix} \frac{\exp 3}{\exp 3 + \exp 1 + \exp 2} & \frac{\exp 1}{\exp 1 + \exp 5 + \exp 6} & \frac{\exp 4}{\exp 4 + \exp 9 + \exp 5} \\ \frac{\exp 1}{\exp 3 + \exp 1 + \exp 2} & \frac{\exp 5}{\exp 1 + \exp 5 + \exp 6} & \frac{\exp 9}{\exp 4 + \exp 9 + \exp 5} \\ \frac{\exp 2}{\exp 3 + \exp 1 + \exp 2} & \frac{\exp 6}{\exp 1 + \exp 5 + \exp 6} & \frac{\exp 5}{\exp 4 + \exp 9 + \exp 5} \end{bmatrix} \overset{\text{width}}{}$$

$$\underset{\mathsf{width}}{\operatorname{softmax}} A = \mathsf{height} \begin{bmatrix} \frac{\exp 3}{\exp 3 + \exp 1 + \exp 4} & \frac{\exp 1}{\exp 3 + \exp 1 + \exp 4} & \frac{\exp 4}{\exp 3 + \exp 1 + \exp 4} \\ \frac{\exp 1}{\exp 1 + \exp 5 + \exp 9} & \frac{\exp 5}{\exp 1 + \exp 5 + \exp 9} & \frac{\exp 9}{\exp 1 + \exp 5 + \exp 9} \\ \frac{\exp 2}{\exp 2 + \exp 6 + \exp 5} & \frac{\exp 6}{\exp 2 + \exp 6 + \exp 5} & \frac{\exp 5}{\exp 2 + \exp 6 + \exp 5} \end{bmatrix} \overset{\text{width}}{}$$

### 3.6 Renaming and reshaping

It's often useful to rename an axis (analogous to a transpose operation in standard notation). We can think of this as the lift of a function from vectors to vectors, but with different input and output axes:

$$[-]_{\mathsf{ax} \to \mathsf{ax'}} : \mathbb{R}^{\mathsf{ax}[I]} \to \mathbb{R}^{\mathsf{ax'}[I]}$$
$$[X_{\mathsf{ax} \to \mathsf{ax'}}]_{\mathsf{ax'}(i)} = X_{\mathsf{ax}(i)}$$

For example,

$$A_{\mathsf{height} \to \mathsf{height'}} = \mathsf{height'} \begin{bmatrix} 3 & 1 & 4 \\ 1 & 5 & 9 \\ 2 & 6 & 5 \end{bmatrix} \overset{\text{width}}{}.$$

We can also define notation for reshaping two or more axes into one axis:

$$A_{(\mathsf{height}, \mathsf{width}) \to \mathsf{layer}} = \begin{bmatrix} 3 & 1 & 4 & 1 & 5 & 9 & 2 & 6 & 5 \end{bmatrix} \overset{\text{layer}}{}$$

The order of elements in the new axis is undefined. Authors who need a particular ordering may write a more specific definition.

### 3.7 Indexing[2]

NumPy and its derivatives provide various ways to recombine elements of a tensor to form a new tensor: integer array indexing, and functions like `numpy.take`, `numpy.take_along_axis`, `torch.index_select`, and `torch.gather`. Using named tensors, we can write nearly all of these operations as lifts of a single function:

$$\underset{\mathsf{ax}}{\operatorname{index}} : \mathbb{R}^{\mathsf{ax}[n]} \times [n] \to \mathbb{R}$$
$$\underset{\mathsf{ax}}{\operatorname{index}}(X, i) = X_{\mathsf{ax}(i)}.$$

For example, suppose we have

$$E \in \mathbb{R}^{\mathsf{vocab}[n] \times \mathsf{emb}}$$
$$i \in [n]$$
$$I \in [n]^{\mathsf{seq}}$$
$$P \in \mathbb{R}^{\mathsf{seq} \times \mathsf{vocab}[n]}$$

---

[2]We are grateful to Tongfei Chen and Chu-Cheng Lin for contributing the original idea behind this section, as well as the example.

Tensor $E$ contains word embeddings for all the words in the vocabulary. Integer $i$ is the numeric identifier of a word, while tensor $I$ is a sequence of numeric identifiers of words. Tensor $P$ contains a sequence of probability distributions over the vocabulary (e.g., the predictions of a language model). Then:

- $\underset{\mathsf{vocab}}{\mathrm{index}}(E, i)$ broadcasts the $\mathsf{emb}$ axis from $E$ to $i$, giving the word embedding of word $i$. This is the same as partial indexing ($E_{\mathsf{vocab}(i)}$).

- $\underset{\mathsf{vocab}}{\mathrm{index}}(E, I)$ also broadcasts the $\mathsf{seq}$ axis from $I$ to $E$, giving a sequence of word embeddings. This is the same as integer array indexing ($E[I]$), `numpy.take(`$E$`, `$I$`, 0)`, or `torch.index_select(`$E$`, 0, `$I$`)`.

- $\underset{\mathsf{vocab}}{\mathrm{index}}(P, I)$ aligns $P$'s and $I$'s $\mathsf{seq}$ axes, giving a sequence of probabilities. This is the same as `numpy.take_along_axis(`$P$`, `$I$`, 0)` or `torch.gather(`$P$`, 0, `$I$`)`.

- If $P$ and $I$ additionally had a $\mathsf{batch}$ axis (before the other axes), then $\underset{\mathsf{vocab}}{\mathrm{index}}(P, I)$ would be the same as `tensorflow.gather(`$P$`, `$I$`, axis=1, batch_dims=1)`.

In NumPy, indexing using two or more integer arrays requires a special definition with some surprising special cases. With named tensors, we simply apply the indexing function twice. For example, if we wanted to get probabilities of words $J$ at a subset $I$ of positions, we could let:

$$
\begin{aligned}
&|\mathsf{seq}| = m \\
&I \in [m]^{\mathsf{subseq}} && \text{positions} \\
&J \in [n]^{\mathsf{subseq}} && \text{numeric identifiers} \\
&S = \underset{\mathsf{vocab}}{\mathrm{index}}(\underset{\mathsf{seq}}{\mathrm{index}}(P, I), J) \in \mathbb{R}^{\mathsf{subseq}}
\end{aligned}
$$

so that

$$
S_{\mathsf{subseq}(k)} = P_{\mathsf{seq}(I_{\mathsf{subseq}(k)}), \mathsf{vocab}(I_{\mathsf{subseq}(k)})}.
$$

# 4 Worked Examples: Neural Networks

In this section we give a series of worked examples illustrating how standard neural network model components can be written using named tensors. Appendix A builds some of these components into complete specifications of the Transformer and LeNet.

## 4.1 Feedforward neural networks

A multi-layer, feedforward neural network with different-sized layers can be written as:

$$
\begin{aligned}
&X^0 \in \mathbb{R}^{\mathsf{input}} \\
&X^1 = \sigma(W^1 \underset{\mathsf{input}}{\odot} X^0 + b^1) && W^1 \in \mathbb{R}^{\mathsf{hidden}_1 \times \mathsf{input}} && b^1 \in \mathbb{R}^{\mathsf{hidden}_1} \\
&X^2 = \sigma(W^2 \underset{\mathsf{hidden}_1}{\odot} X^1 + b^2) && W^2 \in \mathbb{R}^{\mathsf{hidden}_2 \times \mathsf{hidden}_1} && b^2 \in \mathbb{R}^{\mathsf{hidden}_2} \\
&X^3 = \sigma(W^3 \underset{\mathsf{hidden}_2}{\odot} X^2 + b^3) && W^3 \in \mathbb{R}^{\mathsf{out} \times \mathsf{hidden}_2} && b^3 \in \mathbb{R}^{\mathsf{out}}
\end{aligned}
$$

The layer sizes can be specified by writing $|\mathsf{hidden}_1| = n_1$, etc. As noted above, $\sigma$ is applied elementwise and does not require additional annotation.

Alternatively, the layer equation can be abstracted by defining:

$$\text{FullConn}^l(x) = \sigma\left(W^l \underset{\text{layer}}{\odot} x + b^l\right)_{\text{layer}' \to \text{layer}}$$

where

$$W^l \in \mathbb{R}^{\text{layer}'[n_l] \times \text{layer}[n_{l-1}]}$$

$$b^l \in \mathbb{R}^{\text{layer}'[n_l]}.$$

The function $\text{FullConn}^l$ encapsulates both the equation for layer $l$ as well as its parameters $W^l, b^l$ (analogous to what TensorFlow and PyTorch call *modules*). Since we chose to use the same axis name layer for all the layers (with different sizes $n_l$), $\text{FullConn}^l$ temporarily computes its output over axis layer', then renames it back to layer. The network can be defined like this:

$$X^0 \in \mathbb{R}^{\text{layer}[n_0]}$$

$$X^1 = \text{FullConn}^1(X^0)$$

$$X^2 = \text{FullConn}^2(X^1)$$

$$X^3 = \text{FullConn}^3(X^2).$$

## 4.2 Recurrent neural networks

As a second example, we consider a simple (Elman) RNN. This model is similar to the feedforward network, except that the number of timesteps is variable and parameters are shared over time. At each time step, it produces a tensor with a new axis hidden' which is then renamed hidden for the next step in the recursion.

$$x^t \in \mathbb{R}^{\text{input}} \qquad\qquad t = 1, \ldots, n$$

$$W^{\text{h}} \in \mathbb{R}^{\text{hidden} \times \text{hidden}'} \qquad\qquad |\text{hidden}| = |\text{hidden}'|$$

$$W^{\text{i}} \in \mathbb{R}^{\text{input} \times \text{hidden}'}$$

$$b \in \mathbb{R}^{\text{hidden}'}$$

$$h^0 \in \mathbb{R}^{\text{hidden}}$$

$$h^t = \sigma\left(W^{\text{h}} \underset{\text{hidden}}{\odot} h^{t-1} + W^{\text{i}} \underset{\text{input}}{\odot} x^t + b\right)_{\text{hidden}' \to \text{hidden}} \qquad\qquad t = 1, \ldots, n$$

## 4.3 Attention

In the introduction (§1), we described difficulties in interpreting the equation for attention as used with Transformers (Vaswani et al., 2017). In our notation, it looks like this:

$$\text{Attention} \colon \mathbb{R}^{\text{key}} \times \mathbb{R}^{\text{seq} \times \text{key}} \times \mathbb{R}^{\text{seq} \times \text{val}} \to \mathbb{R}^{\text{val}} \tag{6}$$

$$\text{Attention}(Q, K, V) = \left(\underset{\text{seq}}{\text{softmax}} \frac{Q \underset{\text{key}}{\odot} K}{\sqrt{|\text{key}|}}\right) \underset{\text{seq}}{\odot} V. \tag{7}$$

This definition takes a single query $Q$ vector and returns a single result vector (and actually could be further reduced to a scalar values as val is not strictly necessary). To apply to a sequence, we can give $Q$ a seq' axis, and the function will compute an output sequence. Providing $Q$, $K$, and $V$ with a heads axis lifts the function to compute multiple attention heads.

For Transformers we often need to apply a mask to ensure attention is only applied to valid keys (e.g. for causal language models). We can modify the equation to include this mask:

$$\text{Attention} \colon \mathbb{R}^{\text{key}} \times \mathbb{R}^{\text{seq} \times \text{key}} \times \mathbb{R}^{\text{seq} \times \text{val}} \times \mathbb{R}^{\text{seq}} \to \mathbb{R}^{\text{val}}$$

$$\text{Attention}(Q, K, V, M) = \left( \underset{\text{seq}}{\text{softmax}} \left( \frac{Q \underset{\text{key}}{\odot} K}{\sqrt{|\text{key}|}} + M \right) \right) \underset{\text{seq}}{\odot} V.$$

Appendix A.1 includes a full specification of the complete Transformer model using the named tensor notation.

### 4.4 Convolution

Standard neural network convolutions can be specified by "unrolling" a vector and then applying a standard dot product. We define an axis-parameterized unrolling function that converts a one-axis tensor to a sequence of kernel sized vectors:

$$\underset{\substack{\text{seq} \\ \text{kernel}}}{\text{unroll}} : \mathbb{R}^{\text{seq}[n]} \to \mathbb{R}^{\text{seq}[n-|\text{kernel}|+1], \text{kernel}}$$

$$\underset{\substack{\text{seq} \\ \text{kernel}}}{\text{unroll}} X = Y, \text{ where}$$

$$Y_{\text{seq}(i), \text{kernel}(j)} = X_{\text{seq}(i+j-1)}.$$

A 1d convolution with input channels chans and output channels chans$'$ consists of unrolling along the seq axis and then taking a dot product:

$$\text{Conv1d} : \mathbb{R}^{\text{chans} \times \text{seq}[n]} \to \mathbb{R}^{\text{chans}' \times \text{seq}[n']}$$

$$\text{Conv1d}(X; W, b) = W \underset{\substack{\text{chans} \\ \text{kernel}}}{\odot} \underset{\substack{\text{seq} \\ \text{kernel}}}{\text{unroll}} X + b$$

where

$$W \in \mathbb{R}^{\text{chans}' \times \text{chans} \times \text{kernel}}$$

$$b \in \mathbb{R}^{\text{chans}'}$$

Unrolling easily generalizes to higher-dimensional convolutions:

$$\text{Conv2d} : \mathbb{R}^{\text{chans} \times \text{height}[h] \times \text{width}[w]} \to \mathbb{R}^{\text{chans}' \times \text{height}[h'] \times \text{width}[w']}$$

$$\text{Conv2d}(X; W, b) = W \underset{\substack{\text{chans} \\ \text{kh,kw}}}{\odot} \underset{\substack{\text{height} \\ \text{kh}}}{\text{unroll}} \underset{\substack{\text{width} \\ \text{kw}}}{\text{unroll}} X + b$$

where

$$W \in \mathbb{R}^{\text{chans}' \times \text{chans} \times \text{kh} \times \text{kw}}$$

$$b \in \mathbb{R}^{\text{chans}'}.$$

### 4.5 Pooling

Pooling is similar to convolutions. We first define a function to partition a tensor into windows.

$$\underset{\text{seq,kernel}}{\text{pool}} : \mathbb{R}^{\text{seq}[n]} \to \mathbb{R}^{\text{seq}[n/|\text{kernel}|], \text{kernel}}$$

$$\underset{\text{seq,kernel}}{\text{pool}} X = Y, \text{ where}$$

$$Y_{\text{seq}(i), \text{kernel}(j)} = X_{\text{seq}((i-1) \cdot |\text{kernel}| + j)}.$$

Then we can define aggregations over kernel. We define max-pooling as:

$$\text{MaxPool1d}_k \colon \mathbb{R}^{\mathsf{seq}[n]} \to \mathbb{R}^{\mathsf{seq}[n/k]}$$

$$\text{MaxPool1d}_k(X) = \max_{\mathsf{kernel}} \underset{\mathsf{seq},\mathsf{kernel}}{\text{pool}} X$$

$$|\mathsf{kernel}| = k$$

$$\text{MaxPool2d}_{kh,kw} \colon \mathbb{R}^{\mathsf{height}[h] \times \mathsf{width}[w]} \to \mathbb{R}^{\mathsf{height}[h/kh] \times \mathsf{width}[w/kw]}$$

$$\text{MaxPool2d}_{kh,kw}(X) = \max_{\mathsf{kh},\mathsf{kw}} \underset{\mathsf{height},\mathsf{kh}}{\text{pool}} \underset{\mathsf{width},\mathsf{kw}}{\text{pool}} X$$

$$|\mathsf{kh}| = kh$$

$$|\mathsf{kw}| = kw.$$

## 4.6 Normalization layers

Normalization layers are used in all large-scale deep learning models, with different architectures requiring different types of normalization. However, despite their importance, the differences between them are often not clearly communicated. For example, the PyTorch documentation (PyTorch Contributors, 2022) describes all of them using the same equation (where $\epsilon > 0$ is a small constant for numerical stability):

$$Y = \frac{X - \text{mean}(X)}{\sqrt{\text{var}(X) + \epsilon}} \odot \gamma + \beta$$

Wu & He (2018) give essentially the same equation and explain the differences using a combination of equations, words, and pictures. But they do not capture differences in $\gamma$ and $\beta$ among different normalization layers.

Critically, the layers do differ by which axes are *standardized* as well as their parameters. We define a single named standardization function as:

$$\underset{\mathsf{ax}}{\text{standardize}} \colon \mathbb{R}^{\mathsf{ax}} \to \mathbb{R}^{\mathsf{ax}}$$

$$\underset{\mathsf{ax}}{\text{standardize}}(X) = \frac{X - \underset{\mathsf{ax}}{\text{mean}}(X)}{\sqrt{\underset{\mathsf{ax}}{\text{var}}(X) + \epsilon}}$$

Then, we can define the three kinds of normalization layers, all with type $\mathbb{R}^{\mathsf{batch} \times \mathsf{chans} \times \mathsf{layer}} \to \mathbb{R}^{\mathsf{batch} \times \mathsf{chans} \times \mathsf{layer}}$. While superficially similar, these functions differ in their standardized axes and their parameter shape.

$$\text{BatchNorm}(X; \gamma, \beta) = \underset{\mathsf{batch},\mathsf{layer}}{\text{standardize}}(X) \odot \gamma + \beta \qquad \gamma, \beta \in \mathbb{R}^{\mathsf{chans}}$$

$$\text{InstanceNorm}(X; \gamma, \beta) = \underset{\mathsf{layer}}{\text{standardize}}(X) \odot \gamma + \beta \qquad \gamma, \beta \in \mathbb{R}^{\mathsf{chans}}$$

$$\text{LayerNorm}(X; \gamma, \beta) = \underset{\mathsf{layer},\mathsf{chans}}{\text{standardize}}(X) \odot \gamma + \beta \qquad \gamma, \beta \in \mathbb{R}^{\mathsf{chans},\mathsf{layer}}$$

# 5 Differential Calculus

In many machine learning applications, we need to compute derivatives of functions from tensors to tensors. In standard vector/matrix notation, this can become complicated. For example, if $f$ maps from vectors to vectors, then the partial derivatives of $f$ form a matrix (the Jacobian). It has an "input" axis for the directions in which $X$ could change, and an "output" axis for the directions in which $f(X)$ could change. But there are conflicting conventions about whether the first axis is the input axis ("denominator layout") or the output axis ("numerator layout"). The derivative of a function from vectors to matrices or matrices to vectors cannot be represented as a matrix at all, so one must resort to flattening the matrices into vectors.

With non-named tensor index notation, taking derivatives is not difficult (Laue et al., 2018), but again a convention must be adopted that the input axes come after the output axes, separated by a comma.

With named tensors, axes are not ordered, so we don't need to remember whether the input or output axes come first. But we do need to ensure that the input and output axes have different names.

### 5.1 Definition

**Definition 7.** Let $\mathcal{S} = \mathsf{ax}_1 \times \cdots \times \mathsf{ax}_r$ be a shape. Then we write $\mathcal{S}^* = \mathsf{ax}_1^* \times \cdots \times \mathsf{ax}_r^*$, and if $s = \{\mathsf{ax}_1(i_1), \ldots \mathsf{ax}_r(i_r)\} \in \operatorname{rec} \mathcal{S}$, then we write $s^* = \{\mathsf{ax}_1^*(i_1), \ldots \mathsf{ax}_r^*(i_r)\}$. Furthermore, if $X \in \mathbb{R}^{\mathcal{S}}$ then we write $X^* = X_{\mathcal{S} \to \mathcal{S}^*}$.

**Definition 8.** Let $f \colon \mathbb{R}^{\mathcal{S}} \to \mathbb{R}^{\mathcal{T}}$. The *derivative* of $f(X)$ with respect to $X^*$ is the tensor such that

$$\frac{\partial f(X)}{\partial X^*} \in \mathbb{R}^{\mathcal{S}^* \cup \mathcal{T}}$$

$$\left[ \frac{\partial f(X)}{\partial X^*} \right]_{s^*,t} = \frac{\partial f(X)_t}{\partial X_s}$$

for all $s \in \operatorname{rec} \mathcal{S}$ and $t \in \operatorname{rec} \mathcal{T}$.

The above definition assumes that $\mathcal{S}^*$ and $\mathcal{T}$ are orthogonal; if not, the axes in $\mathcal{S}$ should be renamed to something else. For example, the second derivative (the Hessian) could be

$$\frac{\partial^2 f(X)}{\partial X^* \partial X^\dagger} \in \mathbb{R}^{\mathcal{S}^* \cup \mathcal{S}^\dagger \cup \mathcal{T}}$$

$$\left[ \frac{\partial^2 f(X)}{\partial X^* \partial X^\dagger} \right]_{r^*, s\dagger, t} = \frac{\partial^2 f(X)_t}{\partial X_r \partial X_s}$$

for all $r, s \in \operatorname{rec} \mathcal{S}$ and $t \in \operatorname{rec} \mathcal{T}$.

### 5.2 Differentials

We could derive rules like the chain rule and the sum and product rules, and use them to compute derivatives; however, ensuring that input and output shapes are orthogonal is inconvenient. Instead, we recommend the method of differentials (Magnus & Neudecker, 1985), which reduces renaming to a minimum.

The first-order Taylor approximation of $f$ around $X \in \mathbb{R}^{\mathcal{S}}$ is

$$f(X + H) \approx f(X) + \frac{\partial f(X)}{\partial X^*} \underset{\mathcal{S}^*}{\odot} H^* \qquad H \in \mathbb{R}^{\mathcal{S}}.$$

The *differential* of $f(X)$ with increment $H$, written $\partial f(X; H)$, is the second term of this approximation; we defer a formal definition to Appendix B.

For example,

- If id is the identity function, then

$$\mathrm{id}(X + H) = X + H$$
$$\partial \mathrm{id}(X; H) = H. \tag{8a}$$

- If $f(X) = X \underset{\mathsf{ax}}{\odot} X$ where $X \in \mathbb{R}^{\mathsf{ax}}$, then

$$f(X + H) = (X + H) \underset{\mathsf{ax}}{\odot} (X + H)$$
$$= X \underset{\mathsf{ax}}{\odot} X + 2X \underset{\mathsf{ax}}{\odot} H + H \underset{\mathsf{ax}}{\odot} H$$
$$\partial f(X; H) = 2X \underset{\mathsf{ax}}{\odot} H. \tag{8b}$$

It's often more convenient to work directly with the expression $X \underset{\text{ax}}{\odot} X$ instead of $f(X)$, and to write $\partial(X \underset{\text{ax}}{\odot} X)$ for $\partial f(X; H)$. Then, since $\partial X = \partial \text{id}(X; H) = H$, we can write Eq. (8b) simply as

$$\partial(X \underset{\text{ax}}{\odot} X) = 2X \underset{\text{ax}}{\odot} \partial X$$

so that the $H$ has beeen "hidden" inside $\partial X$. More generally, we can derive rules like the following:

$$\partial(U + V) = \partial U + \partial V \tag{9a}$$

$$\partial(U \odot V) = U \odot \partial V + V \odot \partial U \tag{9b}$$

$$\partial\left(\frac{U}{V}\right) = \frac{V \odot \partial U - U \odot \partial V}{V^2} \tag{9c}$$

$$\partial \underset{\text{ax}}{\sum} U = \underset{\text{ax}}{\sum} \partial U \tag{9d}$$

$$\partial(U \underset{\text{ax}}{\odot} V) = U \underset{\text{ax}}{\odot} \partial V + V \underset{\text{ax}}{\odot} \partial U \tag{9e}$$

$$\partial U_s = [\partial U]_s \tag{9f}$$

$$\partial U_{\text{ax}\to\text{ax}'} = [\partial U]_{\text{ax}\to\text{ax}'} . \tag{9g}$$

The chain rule for differentials is

$$\partial f(U) = \left.\frac{\partial f(X)}{\partial X^*}\right|_{X=U} \underset{\mathcal{S}^*}{\odot} \partial U_{\mathcal{S}\to\mathcal{S}^*} \qquad\qquad f \colon \mathbb{R}^{\mathcal{S}} \to \mathbb{R}^{\mathcal{T}}. \tag{9h}$$

Recall that $f$ can be lifted to shapes larger than $\mathcal{S}$. In that case, the rule above still applies, but note that the contraction will still be over $\mathcal{S}$. A special case of this is when $\mathcal{S} = \mathcal{T} = \emptyset$:

$$\partial f(U) = \left.\frac{\mathrm{d} f(x)}{\mathrm{d} x}\right|_{x=U} \odot \partial U \qquad\qquad f \colon \mathbb{R} \to \mathbb{R}. \tag{9i}$$

For example, letting $f(x) = \exp x$ gives the rule

$$\partial(\exp U) = \exp U \odot \partial U. \tag{9j}$$

Using these rules we can compute the differential of a wide variety of expressions. For example, the softmax operator:

$$\partial(\underset{\text{ax}}{\text{softmax}} X) \overset{(5a)}{=} \partial\left(\frac{\exp X}{\underset{\text{ax}}{\sum}\exp X}\right)$$

$$\overset{(9c)}{=} \frac{\left(\underset{\text{ax}}{\sum}\exp X\right) \odot \partial(\exp X) - \exp X \odot \partial\left(\underset{\text{ax}}{\sum}\exp X\right)}{\left(\underset{\text{ax}}{\sum}\exp X\right)^2}$$

$$\overset{(9d)}{=} \frac{\left(\underset{\text{ax}}{\sum}\exp X\right) \odot \partial(\exp X) - \exp X \odot \underset{\text{ax}}{\sum}\partial(\exp X)}{\left(\underset{\text{ax}}{\sum}\exp X\right)^2}$$

$$\overset{(9j)}{=} \frac{\left(\underset{\text{ax}}{\sum}\exp X\right) \odot \exp X \odot \partial X - \exp X \odot \underset{\text{ax}}{\sum}(\exp X \odot \partial X)}{\left(\underset{\text{ax}}{\sum}\exp X\right)^2}$$

$$\overset{(2)}{=} \frac{\left(\underset{\text{ax}}{\sum}\exp X\right) \odot \exp X \odot \partial X - \exp X \underset{\text{ax}}{\odot} (\exp X \odot \partial X)}{\left(\underset{\text{ax}}{\sum}\exp X\right)^2}$$

$$= \frac{\exp X}{\sum_{\mathsf{ax}} \exp X} \odot \partial X - \frac{\exp X}{\sum_{\mathsf{ax}} \exp X} \odot \left( \frac{\exp X}{\sum_{\mathsf{ax}} \exp X} \underset{\mathsf{ax}}{\odot} \partial X \right)$$

$$\overset{(5a)}{=} \operatorname*{softmax}_{\mathsf{ax}} X \odot \partial X - \operatorname*{softmax}_{\mathsf{ax}} X \odot \left( \operatorname*{softmax}_{\mathsf{ax}} X \underset{\mathsf{ax}}{\odot} \partial X \right)$$

$$= \operatorname*{softmax}_{\mathsf{ax}} X \odot \left( \partial X - \operatorname*{softmax}_{\mathsf{ax}} X \underset{\mathsf{ax}}{\odot} \partial X \right). \tag{10}$$

We stop when the only differentials left are $\partial X$.

### 5.3 Derivatives via differentials

If we can get $\partial f(X)$ into so-called *canonical form*,

$$\partial f(X) = D \underset{\mathcal{S}^*}{\odot} \partial X^* + \text{const.} \tag{11}$$

where "const." stands for terms not depending on $\partial X$, then by Magnus & Neudecker's first identification theorem (Theorem 1 in Appendix B), we can conclude that

$$\frac{\partial f(X)}{\partial X^*} = D.$$

When trying to get expressions into canonical form, one helpful fact is that renaming can be thought of as contraction with an identity matrix. First we define the identity matrix with shape $\mathsf{ax} \times \mathsf{ax}'$:

$$[I_{\mathsf{ax},\mathsf{ax}'}]_{\mathsf{ax}(i),\mathsf{ax}'(j)} = \begin{cases} 1 & i = j \\ 0 & i \neq j. \end{cases}$$

Then for any tensor $A$ with an axis $\mathsf{ax}$,

$$A_{\mathsf{ax} \to \mathsf{ax}'} = I_{\mathsf{ax},\mathsf{ax}'} \underset{\mathsf{ax}}{\odot} A. \tag{12}$$

More specifically, if $\partial X \in \mathbb{R}^{\mathcal{S}}$, then

$$\partial X = I_{\mathcal{S},\mathcal{S}^*} \underset{\mathcal{S}^*}{\odot} \partial X^* \tag{13}$$

and then Eq. (4) can usually be used to move the $\underset{\mathcal{S}^*}{\odot} \partial X_{\mathcal{S} \to \mathcal{S}^*}$ to the outermost level of the expression.

Above, we found the differential of the softmax function; now let us find its derivative.

$$\partial \left( \operatorname*{softmax}_{\mathsf{ax}} X \right) \overset{(10)}{=} \operatorname*{softmax}_{\mathsf{ax}} X \odot \left( \partial X - \operatorname*{softmax}_{\mathsf{ax}} X \underset{\mathsf{ax}}{\odot} \partial X \right)$$

$$\overset{(13)}{=} \operatorname*{softmax}_{\mathsf{ax}} X \odot \left( I_{\mathsf{ax},\mathsf{ax}^*} \underset{\mathsf{ax}^*}{\odot} \partial X^* - \operatorname*{softmax}_{\mathsf{ax}} X \odot \left( I_{\mathsf{ax},\mathsf{ax}^*} \underset{\mathsf{ax}^*}{\odot} \partial X^* \right) \right)$$

$$\overset{(4)}{=} \left( \operatorname*{softmax}_{\mathsf{ax}} X \odot \left( I_{\mathsf{ax},\mathsf{ax}^*} - \operatorname*{softmax}_{\mathsf{ax}} X \odot I_{\mathsf{ax},\mathsf{ax}^*} \right) \right) \underset{\mathsf{ax}^*}{\odot} \partial X^*$$

$$\overset{(12)}{=} \left( \operatorname*{softmax}_{\mathsf{ax}} X \odot \left( I_{\mathsf{ax},\mathsf{ax}^*} - \left( \operatorname*{softmax}_{\mathsf{ax}} X \right)^* \right) \right) \underset{\mathsf{ax}^*}{\odot} \partial X^*.$$

This is in canonical form, so we have

$$\frac{\partial}{\partial X} \left( \operatorname*{softmax}_{\mathsf{ax}} X \right) = \operatorname*{softmax}_{\mathsf{ax}} X \odot \left( I_{\mathsf{ax},\mathsf{ax}^*} - \left( \operatorname*{softmax}_{\mathsf{ax}} X \right)^* \right). \tag{14}$$

### 5.4   Lifting

Recall that $f^{\mathcal{S}'}$ is the lift of $f \colon \mathbb{R}^{\mathcal{S}} \to \mathbb{R}^{\mathcal{T}}$ with $\mathcal{S}'$, and in most contexts we can simply write $f$ instead of $f^{\mathcal{S}'}$. However, derivatives are one place where some extra caution is in order. To lighten notation, let's write $g$ for the derivative of $f$:

$$g(X) = \frac{\partial f(X)}{\partial X^*}.$$

Recall that the chain rule (9h) works under lifting, so

$$\partial f^{\mathcal{S}'}(X) = g^{\mathcal{S}'}(X) \underset{\mathcal{S}^*}{\odot} \partial X_{\mathcal{S} \to \mathcal{S}^*}.$$

But the contraction is only over $\mathcal{S}^*$, so it would be incorrect to conclude that $\frac{\partial f^{\mathcal{S}'}(X)}{\partial X} = g^{\mathcal{S}'}(X)$. The derivative of a lift is *not* the lift of a derivative. We must rename and contract $\mathcal{S}'$ as well:

$$\partial f^{\mathcal{S}'}(X) \overset{(9h)}{=} g^{\mathcal{S}'}(X) \underset{\mathcal{S}^*}{\odot} \partial X_{\mathcal{S} \to \mathcal{S}^*}$$

$$\overset{(13)}{=} g^{\mathcal{S}'}(X) \underset{\mathcal{S}^*}{\odot} (I_{\mathcal{S}',\mathcal{S}'^*} \underset{\mathcal{S}'^*}{\odot} \partial X_{\mathcal{S} \cup \mathcal{S}' \to (\mathcal{S} \cup \mathcal{S}')^*})$$

$$\overset{(3)}{=} (g^{\mathcal{S}'}(X) \odot I_{\mathcal{S}',\mathcal{S}'^*}) \underset{(\mathcal{S} \cup \mathcal{S}')^*}{\odot} \partial X_{\mathcal{S} \cup \mathcal{S}' \to (\mathcal{S} \cup \mathcal{S}')^*}$$

$$\frac{\partial f^{\mathcal{S}'}(X)}{\partial X^*} = g^{\mathcal{S}'}(X) \odot I_{\mathcal{S}',\mathcal{S}'^*}.$$

In general, then, the derivative of a lift is the lift of the derivative, multiplied by the identity matrix for the new axes. Intuitively, this is because the derivative is a linear transformation—before lifting, a transformation from $\mathcal{S}^*$ to $\mathcal{T}$. When lifting to $\mathcal{S} \cup \mathcal{S}'$, this transformation must also be lifted, which is what multiplication by $I_{\mathcal{S}',\mathcal{S}'^*}$ accomplishes.

### 5.5   Extended example

As a more elaborate example, we find the derivative of self-attention. For brevity, we omit the factor $\frac{1}{\sqrt{|\text{key}|}}$, and we write $\alpha = \underset{\text{seq}}{\text{softmax}}(Q \underset{\text{key}}{\odot} K)$.

$$\partial \mathrm{Att}(Q, K, V) \overset{(7)}{=} \partial(\alpha \underset{\text{seq}}{\odot} V)$$

$$\overset{(9e)}{=} \alpha \underset{\text{seq}}{\odot} \partial V + V \underset{\text{seq}}{\odot} \partial \alpha. \tag{15}$$

Focus first on the first term, which is the only term depending on $\partial V$:

$$\alpha \underset{\text{seq}}{\odot} \partial V \overset{(13)}{=} \alpha \underset{\text{seq}}{\odot} ((I_{\text{seq},\text{seq}^*} \odot I_{\text{val},\text{val}^*}) \underset{\substack{\text{seq}^* \\ \text{val}^*}}{\odot} \partial V^*)$$

$$\overset{(4)}{=} ((\alpha \underset{\text{seq}}{\odot} I_{\text{seq},\text{seq}^*}) \odot I_{\text{val},\text{val}^*}) \underset{\substack{\text{seq}^* \\ \text{val}^*}}{\odot} \partial V^*$$

$$\overset{(12)}{=} (\alpha_{\text{seq} \to \text{seq}^*} \odot I_{\text{val},\text{val}^*}) \underset{\substack{\text{seq}^* \\ \text{val}^*}}{\odot} \partial V^*$$

$$\frac{\partial}{\partial V^*} \mathrm{Att}(Q, K, V) = \alpha_{\text{seq} \to \text{seq}^*} \odot I_{\text{val},\text{val}^*}.$$

Next, focus on the second term of Eq. (15):

$$V \underset{\text{seq}}{\odot} \partial\alpha \overset{(10)}{=} V \underset{\text{seq}}{\odot} (\alpha \odot (\partial(Q \underset{\text{key}}{\odot} K) - \alpha \underset{\text{seq}}{\odot} \partial(Q \underset{\text{key}}{\odot} K)))$$

$$\overset{(9e)}{=} V \underset{\text{seq}}{\odot} (\alpha \odot (Q \underset{\text{key}}{\odot} \partial K + K \underset{\text{key}}{\odot} \partial Q - \alpha \underset{\text{seq}}{\odot} (Q \underset{\text{key}}{\odot} \partial K + K \underset{\text{key}}{\odot} \partial Q))). \tag{16}$$

Keeping only terms depending on $\partial Q$, we get

$$V \underset{\text{seq}}{\odot} (\alpha \odot (K \underset{\text{key}}{\odot} \partial Q - \alpha \underset{\text{seq}}{\odot} (K \underset{\text{key}}{\odot} \partial Q)))$$

$$\overset{(13)}{=} V \underset{\text{seq}}{\odot} (\alpha \odot (K \underset{\text{key}}{\odot} (I_{\text{key,key}^*} \underset{\text{key}^*}{\odot} \partial Q^*) - \alpha \underset{\text{seq}}{\odot} (K \underset{\text{key}}{\odot} (I_{\text{key,key}^*} \underset{\text{key}^*}{\odot} \partial Q^*))))$$

$$\overset{(4)}{=} \left( V \underset{\text{seq}}{\odot} (\alpha \odot (K \underset{\text{key}}{\odot} I_{\text{key,key}^*} - \alpha \underset{\text{seq}}{\odot} (K \underset{\text{key}}{\odot} I_{\text{key,key}^*}))) \right) \underset{\text{key}^*}{\odot} \partial Q^*$$

$$\overset{(4)}{=} \left( V \underset{\text{seq}}{\odot} (\alpha \odot (K_{\text{key} \to \text{key}^*} - \alpha \underset{\text{seq}}{\odot} K_{\text{key} \to \text{key}^*})) \right) \underset{\text{key}^*}{\odot} \partial Q^*$$

$$\frac{\partial}{\partial Q^*} \text{Att}(Q, K, V) = V \underset{\text{seq}}{\odot} (\alpha \odot (K_{\text{key} \to \text{key}^*} - \alpha \underset{\text{seq}}{\odot} K_{\text{key} \to \text{key}^*})).$$

Similarly, keeping only terms from Eq. (16) depending on $\partial K$, we get

$$V \underset{\text{seq}}{\odot} (\alpha \odot (Q \underset{\text{key}}{\odot} \partial K - \alpha \underset{\text{seq}}{\odot} (Q \underset{\text{key}}{\odot} \partial K)))$$

$$= \left( V \underset{\text{seq}}{\odot} (\alpha \odot (Q^* \odot I_{\text{seq,seq}^*}) - \alpha \underset{\text{seq}}{\odot} (Q^* \odot I_{\text{seq,seq}^*}))) \right) \underset{\substack{\text{key}^* \\ \text{seq}^*}}{\odot} \partial K^*$$

$$\frac{\partial}{\partial K^*} \text{Att}(Q, K, V) = V \underset{\text{seq}}{\odot} (\alpha \odot (Q^* \odot I_{\text{seq,seq}^*}) - \alpha \underset{\text{seq}}{\odot} (Q^* \odot I_{\text{seq,seq}^*}))).$$

# 6 Alternatives and Related Work

## 6.1 Index notations

Among alternatives to standard vector and matrix notation, the most common one is index notation as used in physics (Ricci & Levi-Civita, 1900). Related notations are used in other fields as well (Harshman, 2001). In this notation, axes are ordered, and every equation is written in terms of tensor components. If an index appears on both sides of an equation, then the equation must hold for each value of the index, and the Einstein summation convention (Einstein, 1916) is that if an index appears twice on one side and not on the other, there is an implicit summation over that index.

$$\text{Attention} \colon \mathbb{R}^{d_k} \times \mathbb{R}^{n \times d_k} \times \mathbb{R}^{n \times d_v} \to \mathbb{R}^{d_v}$$

$$[\text{Attention}(Q, K, V)]_k = \underset{i}{\text{softmax}} \left( \frac{Q_j K_{ij}}{\sqrt{d_k}} \right) V_{ik}.$$

Because $k$ appears on both sides, the equation must hold over all values of this index. But because $i$ and $j$ occur twice on only the right-hand side, they are both summed over. We would have to define exactly what the $i$ under the softmax means ($i$ is bound inside the softmax and free outside it), and since softmax doesn't distribute over addition, we would need to modify the summation convention so that the summation over $j$ occurs inside the softmax.

Aside from these correctable issues, this notation scales very well to more than two axes and is concise and unambiguous. But it does not solve the main problem we set out to solve, which is that ordered axes force the author and reader to remember the purpose of each axis. The indices do act as symbolic names for axes

(indeed, in *abstract* index notation (Penrose & Rindler, 1984), they really are symbols, not variables), but they are temporary names; they could be totally different in the next equation. It would be up to the author to choose to use consistent names, and to do so correctly.

A second issue is that because it depends on repetition of indices to work, index notation can be more verbose than our notation, particularly for reductions and contractions:

$$C = \max_i A_i \qquad\qquad C = \max_{\mathsf{ax}} A$$

$$C = A_i B_i \qquad\qquad C = A \underset{\mathsf{ax}}{\odot} B.$$

Finally, index notation requires us to write out all indices explicitly. So if we wanted to lift attention to minibatches ($b \in [B]$), multiple heads ($h \in [H]$) and multiple query tokens ($i' \in [n']$), we would write:

$$\text{Attention}\colon \mathbb{R}^{B \times H \times n' \times d_k} \times \mathbb{R}^{B \times H \times n \times d_k} \times \mathbb{R}^{B \times H \times n \times d_v} \to \mathbb{R}^{B \times H \times n' \times d_v}$$

$$[\text{Attention}(Q, K, V)]_{bhi'k} = \operatorname*{softmax}_i \left( \frac{Q_{bhi'j} K_{bhij}}{\sqrt{d_k}} \right) V_{bhik}.$$

We could adopt a convention that lifts a function on tensors to tensors that have extra axes to the *left*, but such conventions tend to lead to messy reordering and squeezing/unsqueezing of axes. Named axes make such conventions unnecessary.

## 6.2 Graphical notations

In the graphical notation of Penrose (1971), a node in the graph stands for a tensor, and its incident edges stand for its indices. The edges are ordered from left to right. An edge connecting two nodes denotes contraction. The notation of Alsberg (1997) is similar, except that edges are named, not ordered.

Graphs are commonly used in machine learning for representing probability models (Koller & Friedman, 2009). A node in the graph stands for a random variable, and an edge or hyperedge stands for a dependency between variables. If random variables have finite domains, then a (hyper)edge with $r$ endpoints can be thought of as an $r$-th order tensor. A graph can then be thought of as a product and contraction. Extensions that allow for a choice between two subgraphs (e.g., Minka & Winn, 2008) can be thought of as addition.

Our assessment of graphical notations like these is that, on the positive side, they have obvious value for visualization, and they at least have the potential to represent indices in a purely unordered way. On the negative side, these notations seem best suited for representing linear functions, and even for this purpose, some other practical considerations are that drawing pictures requires more effort from the author, and that pictures will have a less transparent relationship with their implementation in most programming languages.

## 6.3 Relational algebra

In relational algebra (Codd, 1970), the basic objects are sets of $r$-tuples, which could be thought of as tensors of order $r$ with Boolean-valued entries. In the original formulation, the members of the tuples, which correspond to axes, were both named *and* ordered, although later definitions (e.g. Pirotte, 1982) made them unordered.

Probabilistic variants of relational algebra also exist (e.g. Dey & Sarkar, 1996; Fuhr & Rölleke, 1997), whose relations would correspond to tensors of probabilities.

While relational algebra and tensor notations are designed for totally different purposes, the notation of relational algebra generally has a similar flavor to ours (for example, our contraction operator is similar to the $\bowtie$ operator, and our renaming operator is the same as the $\rho$ operator).

### 6.4 Programming languages

One response to the notation presented here, as well as the alternative notations mentioned in this section, is that research papers in machine learning should simply present models as code rather than equations. But we argue that a model's mathematical specification should abstract away from details of its implementation.

Conceptually, it is important to have a distinct specification to define what makes an implementation (both the original implementation and any reimplementations) correct or incorrect. If the implementation is its own specification, it cannot be correct or incorrect; it will be "not even wrong."

Practically, abstracting away from implementation is important because we do not want the interpretation of research papers to be subject to differences across programming languages and libraries, or versions thereof. For example, PyTorch's `Dropout2d` on order-3 tensors has one behavior in versions 1.11 and 1.13, but another behavior in 1.10, 1.12, and future versions. It would be problematic for correct understanding of a paper to depend on such differences.

## 7 Conclusions

Named tensor notation is a system of formal notation for representing operations between tensors in a non-ambiguous way while remaining intuitive for practitioners. The system is motivated by challenges that arise from taking notation designed for applied linear algebra and using it for representing neural networks, as demonstrated through examples of canonical deep-learning components such as attention and layer normalization. However, named tensors are not limited to specifying neural networks. We have also explained how to integrate our notation with Magnus & Neudecker (1985)'s method of differentials for matrix calculus. While there are other conventions that such as index notation that have some usage in the machine learning community, these conventions either lack the conciseness of named tensors or are not well-suited to non-linear operations. For these reasons, we encourage members of the machine learning community to try out named tensor notation for teaching, research, and software documentation.

## Acknowledgements

We would like to thank Ekin Akyürek, Justin Bayer, Tongfei Chen, Chu-Cheng Lin, Colin McDonald, Adam Poliak, Matt Post, Chung-chieh Shan, Nishant Sinha, and Yee Whye Teh for their input to this document (or the ideas in it). We also thank the anonymous TMLR reviewers for their feedback, which substantially improved the quality of the paper, especially Section 5.

This material is based upon work supported by the National Science Foundation under Grants No. CCF-2019291 and DMS-2134157, as well as a Sloan Fellowship, Simons Investigator Fellowship, DARPA grant W911NF2010021, and DOE grant DE-SC0022199. Any opinions, findings, and conclusions or recommendations expressed in this material are those of the authors and do not necessarily reflect the views of the funding agencies.

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

## A  Extended Examples

### A.1  Transformer

We define a Transformer used autoregressively as a language model. The input is a sequence of one-hot vectors, from which we compute word embeddings and positional encodings:

$$I \in \{0,1\}^{\mathsf{seq} \times \mathsf{vocab}} \qquad\qquad \sum_{\mathsf{vocab}} I = 1$$

$$W = (E \underset{\mathsf{vocab}}{\odot} I)\sqrt{|\mathsf{layer}|} \qquad\qquad E \in \mathbb{R}^{\mathsf{vocab} \times \mathsf{layer}}$$

$$P \in \mathbb{R}^{\mathsf{seq} \times \mathsf{layer}}$$

$$P_{\mathsf{seq}(p),\mathsf{layer}(i)} = \begin{cases} \sin((p-1)/10000^{(i-1)/|\mathsf{layer}|}) & i \text{ odd} \\ \cos((p-1)/10000^{(i-2)/|\mathsf{layer}|}) & i \text{ even.} \end{cases}$$

Then we use $L$ layers of self-attention and feed-forward neural networks:

$$X^0 = W + P$$
$$T^1 = \mathrm{LayerNorm}^1(\mathrm{SelfAtt}^1(X^0) + X^0)$$
$$X^1 = \mathrm{LayerNorm}^{1'}(\mathrm{FFN}^1(T^1) + T^1)$$
$$\vdots$$
$$T^L = \mathrm{LayerNorm}^L(\mathrm{SelfAtt}^L(X^{L-1}) + X^{L-1})$$
$$X^L = \mathrm{LayerNorm}^{L'}(\mathrm{FFN}^L(T^L) + T^L)$$
$$O = \mathrm{softmax}(E \underset{\mathsf{vocab}}{\odot} \underset{\mathsf{layer}}{X^L})$$

where LayerNorm, SelfAtt and FFN are defined below.

Layer normalization ($l = 1, 1', \ldots, L, L'$):

$$\mathrm{LayerNorm}^l \colon \mathbb{R}^{\mathsf{layer}} \to \mathbb{R}^{\mathsf{layer}}$$
$$\mathrm{LayerNorm}^l(X) = \underset{\mathsf{layer}}{\mathrm{standardize}}(X) \odot \gamma^l + \beta^l$$

$$\beta^l, \gamma^l \in \mathbb{R}^{\mathsf{layer}}$$

We defined attention in §4.3; the Transformer uses multi-head self-attention, in which queries, keys, and values are all computed from the same sequence.

$$\mathrm{SelfAtt}^l \colon \mathbb{R}^{\mathsf{seq}\times\mathsf{layer}} \to \mathbb{R}^{\mathsf{seq}\times\mathsf{layer}}$$

$$\mathrm{SelfAtt}^l(X) = Y$$

where

$$|\mathsf{seq}| = |\mathsf{seq}'|$$

$$|\mathsf{key}| = |\mathsf{val}| = |\mathsf{layer}|/|\mathsf{heads}|$$

$$Q = W^{l,Q} \underset{\mathsf{layer}}{\odot} X_{\mathsf{seq}\to\mathsf{seq}'} \qquad\qquad W^{l,Q} \in \mathbb{R}^{\mathsf{heads}\times\mathsf{layer}\times\mathsf{key}}$$

$$K = W^{l,K} \underset{\mathsf{layer}}{\odot} X \qquad\qquad W^{l,K} \in \mathbb{R}^{\mathsf{heads}\times\mathsf{layer}\times\mathsf{key}}$$

$$V = W^{l,V} \underset{\mathsf{layer}}{\odot} X \qquad\qquad W^{l,V} \in \mathbb{R}^{\mathsf{heads}\times\mathsf{layer}\times\mathsf{val}}$$

$$M \in \mathbb{R}^{\mathsf{seq}\times\mathsf{seq}'}$$

$$M_{\mathsf{seq}(i),\mathsf{seq}'(j)} = \begin{cases} 0 & i \le j \\ -\infty & \text{otherwise} \end{cases}$$

$$Y = W^{l,O} \underset{\substack{\mathsf{heads}\\\mathsf{val}}}{\odot} \mathrm{Attention}(Q,K,V,M)_{\mathsf{seq}'\to\mathsf{seq}} \qquad W^{l,O} \in \mathbb{R}^{\mathsf{heads}\times\mathsf{val}\times\mathsf{layer}}$$

Feedforward neural networks:

$$\mathrm{FFN}^l \colon \mathbb{R}^{\mathsf{layer}} \to \mathbb{R}^{\mathsf{layer}}$$

$$\mathrm{FFN}^l(X) = X^2$$

where

$$X^1 = \mathrm{relu}(W^{l,1} \underset{\mathsf{layer}}{\odot} X + b^{l,1}) \qquad W^{l,1} \in \mathbb{R}^{\mathsf{hidden}\times\mathsf{layer}} \qquad b^{l,1} \in \mathbb{R}^{\mathsf{hidden}}$$

$$X^2 = W^{l,2} \underset{\mathsf{hidden}}{\odot} X^1 + b^{l,2} \qquad W^{l,2} \in \mathbb{R}^{\mathsf{layer}\times\mathsf{hidden}} \qquad b^{l,2} \in \mathbb{R}^{\mathsf{hidden}}.$$

## A.2 LeNet

$$X^0 \in \mathbb{R}^{\mathsf{batch}\times\mathsf{chans}[c_0]\times\mathsf{height}\times\mathsf{width}}$$

$$T^1 = \mathrm{relu}(\mathrm{Conv}^1(X^0))$$

$$X^1 = \mathrm{MaxPool}^1(T^1)$$

$$T^2 = \mathrm{relu}(\mathrm{Conv}^2(X^1))$$

$$X^2 = \mathrm{MaxPool}^2(T^2)_{(\mathsf{height},\mathsf{width},\mathsf{chans})\to\mathsf{layer}}$$

$$X^3 = \mathrm{relu}(W^3 \underset{\mathsf{layer}}{\odot} X^2 + b^3) \qquad W^3 \in \mathbb{R}^{\mathsf{hidden}\times\mathsf{layer}} \qquad b^3 \in \mathbb{R}^{\mathsf{hidden}}$$

$$O = \mathrm{softmax}(W^4 \underset{\mathsf{hidden}}{\underset{\mathsf{classes}}{\odot}} X^3 + b^4) \qquad W^4 \in \mathbb{R}^{\mathsf{classes}\times\mathsf{hidden}} \qquad b^4 \in \mathbb{R}^{\mathsf{classes}}$$

As an alternative to the flattening operation in the equation for $X^2$, we could have written

$$X^2 = \mathrm{MaxPool}^2(T^2)$$

$$X^3 = \text{relu}(W^3 \underset{\substack{\text{height}\\\text{width}\\\text{chans}}}{\odot} X^2 + b^3) \qquad\qquad W^3 \in \mathbb{R}^{\text{hidden}\times\text{height}\times\text{width}\times\text{chans}}.$$

The convolution and pooling operations are defined as follows:

$$\text{Conv}^l(X) = \text{Conv2d}(X; W^l, b^l)_{\text{chans}'\to\text{chans}}$$

where

$$W^l \in \mathbb{R}^{\text{chans}'[c_l]\times\text{chans}[c_{l-1}]\times\text{kh}[kh_l]\times\text{kw}[kw_l]}$$
$$b^l \in \mathbb{R}^{\text{chans}'[c_l]}$$

and

$$\text{MaxPool}^l(X) = \text{MaxPool2d}_{ph^l, ph^l}(X).$$

## B  Differentiation: Formal Definitions

The following definition and theorem come directly from the paper by Magnus & Neudecker (1985), but generalized to named tensors.

For any $X \in \mathbb{R}^{\mathcal{S}}$, we write $\|X\| = \underset{\mathcal{S}}{\text{norm}}\, X$.

**Definition 9.** Let $f\colon S \to \mathbb{R}^{\mathcal{T}}$ where $S \subseteq \mathbb{R}^{\mathcal{S}}$. Let $A$ be an interior point of $\mathbb{R}^{\mathcal{S}}$, that is, for some $r > 0$, $B(A; r) = \{X \mid \|X - A\| < r\} \subseteq S$. If there is a tensor $D(A) \in \mathbb{R}^{\mathcal{S}^* \cup \mathcal{T}}$ and $R(A, H) \in \mathbb{R}^{\mathcal{T}}$ such that

$$f(A + H) = f(A) + D(A) \underset{\mathcal{S}^*}{\odot} H_{\mathcal{S}\to\mathcal{S}^*} + R(A, H)$$

for all $H \in \mathbb{R}^{\mathcal{S}}$ with $\|H\| < r$, and

$$\lim_{H \to \mathbf{0}} \frac{R(A, H)}{\|H\|} = \mathbf{0},$$

then $f$ is said to be *differentiable* at $A$; the tensor

$$\partial f(A; H) = D(A) \underset{\mathcal{S}^*}{\odot} H_{\mathcal{S}\to\mathcal{S}^*}$$

is then called the *(first) differential of $f$ at $A$ with increment $H$*.

Magnus & Neudecker give their (first) identification theorem twice, once for vector-to-vector functions and once for matrix-to-matrix functions (but omitting vector-to-matrix and matrix-to-vector functions). Here, we only need one version, which works for functions from tensors to tensors of any shape.

**Theorem 1.** *Let $f\colon S \to \mathbb{R}^{\mathcal{T}}$, where $S \subseteq \mathbb{R}^{\mathcal{S}}$, be differentiable at $A \in S$. Let $D(X) \in \mathbb{R}^{\mathcal{S}^* \cup \mathcal{T}}$. Then*

$$\text{for all } H,\ \partial f(A; H) = D(X) \underset{\mathcal{S}^*}{\odot} H_{\mathcal{S}\to\mathcal{S}^*} \qquad \text{iff} \qquad \left.\frac{\partial f(X)}{\partial X^*}\right|_{X=A} = D(X).$$

## C  LaTeX Macros

Many of the LaTeX macros used in this document are available in the style file `namedtensor.sty`, available on CTAN at `https://ctan.org/pkg/namedtensor`. To use it, put

```
\usepackage{namedtensor}
```

in the preamble of your LaTeX source file (after `\documentclass{article}` but before `\begin{document}`).

We write axis names in sans-serif font. To make this easier, `\name{ax}` prints an axis name (like this: ax), and `\ndef{\ax}{ax}` defines a macro `\ax` that does the same.

- Binary operators
  - Use `A \ndot{\ax} B` for contraction: $A \underset{\mathsf{ax}}{\odot} B$. You can use `\\` to stack up several names.
  - In general, you can use `\nbin` to make a new binary operator with a name under it: `A \nbin{\ax}{\star} B` gives you $A \underset{\mathsf{ax}}{\star} B$.

- Functions
  - Use `\nsum{\ax} A` for summation: $\underset{\mathsf{ax}}{\sum} A$.
  - In general, you can use `\nfun` to make a function with a name under it: `\nfun{\ax}{qux} A` gives you $\underset{\mathsf{ax}}{\mathrm{qux}} A$.

