# OpenReview forum: "Named Tensor Notation"
_TMLR — Accepted by TMLR_

### Review · Reviewer_TBKC · 2022-10-07

**Summary Of Contributions:**

This work offers a named tensor notation in which each dimension/axis of a particular tensor is given a name and the names are used to create unambiguous and easily understood tensor operations. This syntax can be used to dramatically improve the readability of machine learning models, as demonstrated by multiple instances. Detailed comparisons between this notation with other existing tensor notations, including Einstein summation notation and the graphical tensor diagrams, are also given.

**Requested Changes:**

Below are my suggestions:

1. The tensor contraction examples shown in the paper all involve 2 tensors/matrices/vectors. I think it would be helpful to give an example showing how to express general tensor contractions with more than 2 tensors with the named notation. For example, the contraction in the einsum notation $d_{i,k,l} = \sum_{j}a_{i,j,k}b_{i,j}c_{i,j,l}$ can be expressed as $d = \sum_{j}(a \odot b \odot c)$ in the named notation.

2. Definition of derivatives under nonorthogonal case is a bit confusing, since it's not clear to me how to use that in the chain rule. For the example in 5.3, authors show how to get $\frac{\partial L}{\partial X}$ based on differential. Is there easy ways one can derive it using chain rule: $\frac{\partial L}{\partial X}=\frac{\partial L}{\partial Y}\frac{\partial Y}{\partial X}$? It's not clear to me how to use the chain rule directly here since index renaming is needed for $\frac{\partial Y}{\partial X}$. I believe more discussions here would be helpful.

3. Minor: at the beginning of page 13: $\frac{\partial L}{\partial X}$ instead of $\frac{\partial f(Y)}{\partial X}$.


**Strengths And Weaknesses:**

Overall, this is a well-motivated and clearly presented study with useful results for researchers in machine learning. Einstein summation notation is the prevalent tensor notation (also called index notation in the paper). This notation has the advantage of clearly expressing all multilinear operations, but it is insufficient for machine learning algorithms because it cannot handle other operations (convolution, pooling, etc.). The named tensor notation, in my opinion, generalizes the index notation and would help scientific writing in the machine learning field.

---

> ### Author Response · Authors · 2022-10-21
> **Thank you for your review**
>
> 1. We will add to Section 3.3 a discussion of the alternative notation $\sum_{\mathsf{ax}} A \odot B$ and its possible benefits, including (as you noted) the fact that it extends to three or more tensors and (as noted at the bottom of page 11) it makes some algebraic manipulations easier.
>
> 2. We will clarify section 5 and in particular explain why we recommend using differentials over the chain rule. In brief, with differentials, dealing with non-orthogonality just requires one renaming, but with the chain rule, non-orthogonality can appear many times, requiring many renamings.
>
> 3. We'll make this correction; thanks!

---

### Review · Reviewer_qUcr · 2022-10-11

**Summary Of Contributions:**

This paper presents a notation for representing functions between multidimensional arrays (tensors), for use in academic papers and presentations. The specific contributions are:

- The authors define a notation for representing tensors in terms of semantically named axes, and operations that act on those tensors, motivated by drawbacks of existing notation used in papers.
- They specify the interpretation of that notation, in particular by interpreting a tensor as a mapping from a set of named indices (a "record") to a field element, and describing the corresponding mappings for each of their operations.
- They show how the notation can be used to write a variety of operations common in neural network architectures.
- They explain how to use this notation to compute Jacobian tensors, and present a set of rules analogous to the scalar differentiation rules.
- They give examples of how to compute Jacobian tensors for a few operations of interest.

**Broader Impact Concerns:**

I do not have any broader impact concerns.

**Requested Changes:**

## Important requested changes
There were some things I found confusing about the current version of the paper, specifically regarding the computation of derivatives. I would like to see these addressed before recommending acceptance.

### Confusing/incomplete definition of differentials (Section 5.2)

I found Section 5.2 to be very unclear, specifically regarding the definition of a differential. Upon first read, I was very confused regarding where $\Delta X$ is defined, what type of object $\partial f(X; \Delta X)$ is, and why it is OK to omit $\Delta X$ in the calculations; I initially thought the differential must be a function taking $\Delta X$ as an argument, but realized this didn't make any sense once I got to the list of rules.

After consulting Magnus & Neudecker (1985), I think that my confusion comes from two things:
- a missing part of the definition of differential. Magnus & Neudecker define "the differential of f(X) at X **with increment $\Delta X$**", where $\Delta X$ is also explicitly named as part of the term, and different increments produce different differentials.
- the current wording that "the differential ... is a tensor ... that linearly approximates f", which makes it sound like the tensor itself is a mapping somehow (since a fixed tensor can't linearly approximate a function).

If my understanding is correct, I would recommend rewording this section to be more explicit about the role of $\Delta X$ in the differential, and explicitly naming it as part of the definition like Magnus & Neudecker do. Perhaps something like

> Intuitively, the differential of $f(X)$ at $X$ with increment $\Delta X$, written $\partial f(X; \Delta X)$, is a tensor with the same shape as $f(X)$, obtained by evaluating a linear approximation of $f$ around $X$ at the position $X + \Delta X$.

or even

> Intuitively, the differential of $f(X)$ at $X$ with increment $\Delta X$, written $\partial f(X; \Delta X)$, is the first-order term of the Taylor series for $f(X)$ around $X$, evaluated at $X + \Delta X$.

It might be even simpler to just write out a formal definition of the differential instead of omitting it; it looks like the definition would only be a few lines long?

Relatedly, I think the sentence "In practice, we can write $\partial f(X; \Delta X)$ simply as $\partial f$" is not quite correct, and should instead be "as $\partial f(X)$", since subsequent uses of $\partial$ always apply to terms, not functions. (I also think "never appears in our calculations" isn't the best justification, and it would be better to say that it's omitted because it can be inferred from context.)

One more minor point on the same topic: it seems that this section is a recommendation / example of how to use the notation to compute derivatives, so I'd suggest wording it as "we can use the method of differentials" or "we propose/recommend using the method of differentials" instead of just saying "we use the method of differentials". At first I thought this section might be some sort of implementation detail of some autodiff system used by the authors, which was confusing.

### Definition of chain rule (Section 5.2)

The current definition of the chain rule is only well defined when $\\mathcal{U}$ and  $\\mathcal{V}$ are orthogonal, but this is not explicitly stated. Moreover, I think it's important to discuss how to do the chain rule when shapes are NOT orthogonal, since this will occur often in practice. My guess is that this will involve a rename of any conflicting axes, something like

$$
\\partial f(U) = \\left[ \\frac{\\partial f(U)_{\\mathcal{U} \to \\mathcal{U}'}}{\\partial U} \\odot_\\mathcal{U} \\partial U \\right]\_{\\mathcal{U}' \to \\mathcal{U} }
$$

perhaps. (Alternatively the rename could be applied to the denominator, I'm not sure which is simpler.)

### Additional details in example derivations (Section 5.3)

I found the examples in 5.3 hard to verify. While each of the steps seemed plausible, it was difficult to determine which of the rules were being applied, and whether they were being applied correctly. As a derivation of the equations themselves, this seems OK, but as an example of how to use the notation and rules, I think it is important to actually show why the steps follow from the stated rules.

One thing I noticed was the combination of the chain rule with the quotient rule in a single step; I think it would be better to split those steps up.

I also noticed that the transformation from $\partial \exp X$ to $\exp X \odot \partial X$ (hidden in the above step) involves an elementwise variant of the chain rule that is never explicitly stated. The chain rule written in section 5.2 would suggest that the differential would be  $\partial \exp X = \exp X \odot_{ax} \partial X$, but this doesn't work, because it reduces away `ax` incorrectly (since, as a vector-to-vector function, $\exp$ doesn't satisfy the unstated orthogonality requirement). It seems important to either explicitly discuss the chain rule for elementwise functions, or show how it follows from the ordinary chain rule after applying all of the necessary renaming.

Additionally, at multiple places throughout the examples, the canonical form of the differential is not actually shown, despite how much 5.2 stresses the importance of the canonical form. I suggest explicitly writing out the canonical form (with a contraction instead of a sum) at all of the places where Theorem 1 is used in sections 5.3 and 5.4.

---

## Other suggestions
I think the paper could be improved by adding some more discussion of these areas.

### Introduction of named tensors (Section 2.0)
I found the introduction of named tensors a bit disorienting. I think part of my confusion comes from the section being written from a perspective of named axes already existing, despite the notation not being defined yet. For instance, it currently reads "If it has ax1[n1], . . . , axr[nr] (with ax1, . . . , axr being distinct names), it can be thought of as a map that takes as input a record {ax1[i1],...,axr[ir]}, with i1 ∈ [n1],...,ir ∈ [nr], and outputs a field element." However, as a reader, I don't yet know what `ax1[n1]` means, or what it means for a named tensor to have it, or what a record is, and only in the following sections do I learn that `ax1` is a variable taking the place of a name, and `n1` is supposed to denote the size of that axis.

I think it would be clearer to briefly define each of the concepts (informally) before using them, and only introduce one at a time. For instance, perhaps something like

> In contrast, we propose named tensor notation, in which each axis has a name that describes it and ensures there is no confusion between axes. We use `ax[n]` to refer to an axis with name `ax` and size `n`, and `ax(i)` to the `i`th element along that axis (called a named index). We can then define a named tensor as a map that takes a *record* and maps it to a field element, where a record is ...

Another confusing aspect was that the record notation is inconsistent with the later definition, it should be $\\{ax_1(i_1),...,ax_r(i_r)\\}$. (I also mention this in the "Nits" section below.)

### Elementwise operations as a special case of ordinary broadcasting (Section 3.2)
It seems to me that the rules for broadcasting elementwise operations follow directly from the rules for arbitrary functions, if you view elementwise operations as operations on tensors with the empty shape (as mentioned at the end of section 2.2). This seems pretty cool, and might be worth an explicit mention.

### Discussion of tensor-to-tensor operations with multiple (distinct) axes
All of the examples in section 3 appear to treat input axes identically, e.g. summing over two axes at once is the same as flattening a sum over all elements of a reshaped axis. However, later examples use functions (such as `unroll`) which take multiple axes as arguments and apply different operations along each. It might be worthwile to add a brief mention of this (perhaps in section 3.4), since it seems like a strength of the notation that such functions are easy to express.

### Discussion of gather/scatter-type operations

Does this notation support "scatter" / "gather" operations, where one tensor is used to index into another? How would such operations be represented? It would be interesting to add a discussion of those, since those are fairly common when implementing ML algorithms.


### Motivation for differentiation section
Section 5 ("Differentiation") seems a bit out-of-place currently, as it jumps directly into definitions and rules without explaining why. Is this section supposed to be another example of why the notation is useful, by showing that it aids computations? Or are the derivative rules intended to be part of the notation? It would be useful to start this section with some sort of statement summarizing what this section does and how it relates to the remainder of the paper.

### Example of applying chain rule with non-orthogonal axes
Building on my earlier point about the chain rule with non-orthogonal names, I think it would be useful to see an example of a derivation of a differential that involves a non-trivial chain rule application for a function that preserves some axes in its output. For instance, a computation of something like $\\partial ( softmax_{seq} ( Q \\odot_{key} V ))$. Right now the rules don't clearly explain what to do in this situation, as far as I can tell.

### Discussion of other alternative notations

Two other notation options which might be worth discussing:

- A version of index notation that doesn't include the Einstein summation convention, and requires all summation to be written out (a bit less ambiguous than including Einstein summation, but more verbose)

- Just writing out everything in code, e.g. in numpy/pytorch/jax

Not that these are particularly compelling alternatives, but I've seen them used in practice.

### LaTeX definitions for notation

Since this paper is recommending a new type of notation for paper-writing specifically, it would be convenient to include a reference for how to typeset this notation in LaTeX. This might help standardize some of the details (e.g. the centering of subscripts below $\\odot$ and the sans-serif notation for names).

---

## Nits / typos / minor suggestions
I noticed a few minor things while reading through the paper:

- Named tensor software: There have been some newer software implementations of named tensors that aren't mentioned here, including [einops](https://openreview.net/pdf?id=oapKSVM2bcj), [torchdim](https://github.com/facebookresearch/torchdim) and [jax.xmap](https://jax.readthedocs.io/en/latest/notebooks/xmap_tutorial.html). Also, the citation for Dex could be updated to refer to the [conference version](https://dl.acm.org/doi/abs/10.1145/3473593) instead of the workshop paper.

- `Section 2.0`: Inconsistent notation of introductory example. "a map that takes as input a record $\\{ax_1[i_1],...,ax_r[i_r]\\}$" is inconsistent with the notation defined in the next section; it should be $\\{ax_1(i_1),...,ax_r(i_r)\\}$.
  - I found this pretty confusing at first, since it seemed like the $ax_1[i_1]$ notation was being overloaded.

- `Definition 5 (Named tensor operation)`: It seems that your definition of operation is equivalent to a function, and throughout the paper you use "function" and "operation" interchangeably. I wonder if you even need to formally define "operation", or whether "function" would be sufficient?

- `Definition 7 (lifting, binary)`: Slightly odd grammar; perhaps "and **for which** S' ∪ T' is orthogonal to U" would be better

- `Section 3.2`: It would be helpful to restate the definition of $A$ before applying it, or at least refer back to where A was defined (on page 2).

- `Section 3.3`:
  - If "norm" is intended as a convention other authors should use, it might be worth allowing other norms than the L2 norm. (Or, if this is just an example for how to define a reduction, it might be worth stating that explicitly.)
  - Could you include an example of contracting multiple axes at once?

- `Section 3.4`: I was surprised that `argmax` was defined as a one-hot embedding, that seems unusual. Is there not a way to encode the ordinary notion of `argmax` as an integer-valued tensor?

- `Section 4.1`: It's not clear what it means for `FullConn` to "encapsulate" the parameters if the parameters are arguments to `FullConn`. It seems clearer not to have W and b be inputs to `FullConn` if the intent is to have `FullConn` implicitly carry W and b.

- `Section 4.4`: I expected the bias term for convolutions to have a `chans'` axis, is it intentional that it does not?

- `Section 5.4`: Missing word(s): "But although f' can be to X using the usual lifting rules". f' can be what to X?

- `Appendix A.1`: LayerNorm is defined in terms of XNorm, but XNorm is not defined. Perhaps this was supposed to refer to "standardize" from the main paper?



**Strengths And Weaknesses:**

Strengths:

- The proposed notation is quite clean and minimal while also being unambiguous, and I think adopting this notation would lead to more readable ML papers.
- The notation also appears to be very general and expressive, and easy to extend if needed.
- The examples provided by the authors both demonstrate the value of the notation and serve as a reference for future uses of the notation.
- Overall I think this notation will be of interest to members of the TMLR community.

Weaknesses:

- I found some parts of the paper confusing, as discussed in the requested changes section below.
- The definition of differential and corresponding differentiation rules seemed imprecise and not fully specified, which might lead to difficulty using this notation to express derivative computations.
- It's not currently obvious from the paper how to actually typeset this notation (e.g. using LaTeX)

---

> ### Author Response · Authors · 2022-10-21
> **Thank you for your very thoughtful review**
>
> # Derivatives
>
> We will add a bit of motivation at the top of this section to describe its purpose. We will also clarify that the method of differentials is not the only way to compute derivatives with named tensors; it's simply the way that we recommend.
>
> Regarding the definition of differential, thank you for taking the time to refer to Magnus and Neudecker's paper and offering suggestions on how to clarify the definition. We will incorporate your suggestions.
>
> Regarding the chain rule in Section 5.2, you're right that we need to specify that $\mathcal{U}$ and $\mathcal{V}$ are orthogonal. More importantly, we need to explain that, because of lifting, $U$ could have a larger shape than $\mathcal{U}$. In particular, perhaps the most useful instance of this rule is when $\mathcal{U} = \mathcal{V} = \emptyset$ (i.e., $f$ is a scalar-to-scalar function) but $U$ can have any shape. In that case, the rule is $\partial f(U) = f'(U) \odot \partial U$. This is the rule used to derive $\partial(\exp U) = \exp U \odot \partial U$, used in Section 5.3. We will explain this much more clearly in Section 5.2, including the special case of the chain rule for scalar-to-scalar functions and illustrating with $\partial(\exp U)$.
>
> In the examples, we will definitely write out each step and give a justification for each one. We will also make sure that the step just before finding the Jacobian appears exactly in canonical form. Finally, we will add a more involved example; you suggested the softmax of a contraction, but we'll probably go further and show self-attention.
>
> # Other suggestions
>
> - Introduction of named tensors: We agree that the initial mention of named axes became a little bit out of order in the final editing of the draft. We will revise it along the lines of what you have suggested.
>
> - Elementwise operations as a special case of ordinary broadcasting: The end of 3.0 ("then show how this definition leads to many common operations") was intended to communicate that Sections 3.2-3.5 are all special cases of lifting, but we will emphasize this much more clearly.
>
> - Operations with multiple axes treated differently: Sections 4.4 (convolution) and 4.5 (pooling) include examples of such operations.
>
> - Gather/scatter-type operations: We do have a treatment of these types of operations which uses a single operation called "index" that nicely subsumes indexing by an integer array, `take`, `index_select`, `gather`, and `batch_gather`. We omitted this section partly because we thought it was too implementation-oriented, but would be happy to put it back.
>
> - You asked about our representation of argmax: Yes, it's unusual. With the inclusion of "index" it could be more convenient for argmax to return integer indices, but if the argmax is taken over multiple axes, it could get complicated. Of course, authors are always free to choose a different definition of argmax as suits their needs.
>
> - LaTeX definitions: We do provide LaTeX macros, and we omitted the section that describes them, because we thought it might be too nuts-and-bolts for a journal article. However, we would be happy to put this section back into the appendix. We also intend to submit the macros to CTAN.
>
> - While you are right that Definition 5 just defines functions from tensors to tensors, we think it can be useful to give them a special name "named tensor operation", because the lifting convention applies to them and not to ordinary functions, and because we use this term to include unary and binary operators that are not written using typical function notation.
>
> - Other suggestions: We generally agree with your corrections and will be happy to incorporate them into the paper. Thank you!

---

> > ### Comment · Reviewer_qUcr · 2022-11-10
> > **Response to author comments and new revision**
> >
> > I am glad the authors found my review useful. Thank you for adding the new section on the `index` operator, and for clarifying the meaning of differentials.
> >
> > A few more comments related to differentials:
> > - The `*` notation in Definition 7 seems like a convenient way to reason about axis names for derivatives, and overall I like it. However, it seems like it would break if you tried to take higher-order derivatives, because then the shape $\mathcal{T}$ already has `*`s in it and so you might end up with $\mathcal{S}^*$ and $\mathcal{T}$ not being orthogonal.
> >   - This is admittedly a bit nitpicky. If you want to just add a statement that you assume $\mathcal{T}$ doesn't have any `*`s in it that seems OK to me.
> >   - It's possible there's a more elegant notation that would work for higher-order derivatives naturally here, I'm not sure. Maybe something like $\mathcal{S}^{in} \times \mathcal{T}^{out}$, which for higher derivatives would become $\mathcal{U}^{in} \times (\mathcal{S}^{in,out} \times \mathcal{T}^{out,out})$ or something like this? Or introducing subscripts like $\mathcal{U}^{\*_2} \times (\mathcal{S}^{*_1} \times \mathcal{T})$?
> > - In section 5.2, the definition of the first order Taylor expansion uses a strict equality which I don't think is quite correct. I'd suggest using '$\approx$' to make it clear that this is an approximation of $f$.
> > - It seems that the statement about omitting $H$ has been removed, and there doesn't seem to be any other explanation for what the differential notation means when applied to terms. Currently equation (8a) jumps straight into using $\delta U$ notation without defining what that means. As far as I understand it, the key thing that I don't think is very well explained is that you're using $\delta U$ to represent something like $\delta u(X; H)$ where $U = u(X)$ is implicitly a function of $X$ and both $X$ and $H$ are inferred from context. I think it would be worth adding a few sentences introducing this shorthand.
> >
> > Some misc other comments:
> > - There appears to be a missing word in Section 4.3: "This definition takes a single query Q vector **and** returns"
> > - Thanks for including the appendix section on typesetting the notation, and it's good to hear you will submit the macros to CTAN. If you would prefer not to include the "nuts and bolts" in the journal paper, I'd also be fine with just linking to the CTAN package or style files and describing the details there, as long as readers know where to find it.

---

> > > ### Author Response · Authors · 2022-11-12
> > > **More on derivatives**
> > >
> > > Thank you again for your feedback!
> > >
> > > - Regarding second derivatives: In the first revision, we made renaming obligatory and we also made it implicit by writing $\partial f(X) / \partial X$ instead of $\partial f(X) / \partial X_{\mathcal{S}\rightarrow\mathcal{S'}}$. We can back off a little bit and make the renaming explicit, which leaves writers free again to choose whatever naming convention suits them best. In a second derivative, they just have to choose something different for the two input shapes.
> > >
> > > - We will add some explanation about the shift from differentials of functions to differentials of terms.

---

### Review · Reviewer_V2jH · 2022-11-03

**Summary Of Contributions:**

The paper formally introduces the notion of naming tensor axes/dimensions/... and shows how to adapt common tensor operations to this convention. A number of worked examples on common building blocks (MLPs, RNNs, attention, convolutions, ..., up to differentiation) show that standard machine learning tools can be concisely expressed using this notation.

**Broader Impact Concerns:**

I have no concerns regarding the broader impact of this paper.

**Requested Changes:**

* page 1: I found $\mathbb{R}^{\textsf{key}}$ (etc.) surprising, as it is not refering to the dimension of individual keys etc. This aspect becomes clearer in the following, but it would be good to call this out earlier.
* page 2: https://jax.readthedocs.io/en/latest/notebooks/xmap_tutorial.html is another example of an implementation of the proposed notation (which actually discusses HW implementation issues in great detail)
* page 4, def 6/7: I'd recommend to rename $s$ to $s'$ here, as it is an element of $\textnormal{rec}\mathcal{S}'$ (and not the also existent $\textnormal{rec}\mathcal{S}$)
* page 5, Sect. 3.2: please recall for the reader that $A$ was defined in Sect. 2.1. Given that $A$ is also used as a variable in some definitions, it may be easier for the reader if you picked another name for this concrete matrix.
* page 7, Sect. 4.1: "(analogous to what TensorFlow and PyTorch modules)" misses its verb


**Strengths And Weaknesses:**

* (+) the proposed notation is indeed suitable to resolve ambiguities in defining common mechanism.
* (+) the breadth of worked examples shows that the proposed notation is suitable to handle a wide array of use cases.
* (-) there is little novelty here, given that it summarises a trend of (referenced) similar proposals.
* (-) the paper is focused only on notation. However, one reason for choosing to represent operations as (somewhat ambigious and hence error-prone) matrix multiplication is that this closely corresponds to factors influencing hardware performance. Keeping these factors in mind is crucial to enable scale-out and hence important in communicating results.

---

> ### Author Response · Authors · 2022-11-07
> **Thank you for your review**
>
> # Novelty
>
> Regarding novelty, we acknowledge that the paper makes a different sort of novel contribution than typical machine learning papers do. This was one reason we decided to submit to TMLR, which explicitly uses interest rather than novelty as a criterion for acceptance. Vis-a-vis the implementations of named tensors listed at the end of Section 1, we again acknowledge that the idea of naming axes originates with these implementations and not with us. However, this paper is the first attempt, to our knowledge, to design a mathematical notation for named tensors that can be used in papers, so we would like to say more about the relationship between notation and implementation.
>
> First, the genesis of this project was the observation that there is a plethora of libraries that implement named tensors, but people don't seem to be adopting them. We thought that one reason people were still coding with standard vector/matrix data structures could be that the papers they were implementing were also written in standard vector/matrix notation. And one reason papers were using standard vector/matrix notation is that named tensor notation did not yet exist. We wrote this paper to fill this gap.
>
> Second, math and code sometimes have different concerns. For example, standard math notation lacks a way to specify what axis a softmax is performed on, but this has never been a problem for deep learning toolkits. As another example, in code, most people would consider a keyword argument `axis=` to be good because it's descriptive, but in math, this would be unacceptably verbose. So it's not the case that this paper simply summarizes existing proposals; it also addresses concerns that are particular to math that have not been (and could not be) addressed by existing proposals for code.
>
> # Performance considerations
>
> Regarding the observation that the choice of notation (especially the use of matrix multiplies) can have performance implications, we have a few remarks. First (related to the previous section), since our notation is designed for specification, not implementation, it places clear and unambiguous expression of ideas first, with considerations of efficiency being secondary.
>
> Second, vector/matrix notation of course predates the vector and matrix algorithms used on modern hardware, so there is nothing in the notation that intrinsically makes it more efficient than other notations. Tensor contractions can also be implemented efficiently, and can express not only matrix multiplication but operations (e.g., bmm) that are highly parallelizable but inconvenient to express in standard vector/matrix notation. Indeed, we believe that abstracting away unnecessary details (e.g., a total ordering of axes) can make optimizations easier rather than harder. The JAX xmap documentation, linked by you and another reviewer, emphasizes precisely this point.
>
> # Axis names
>
> Regarding our convention of naming axes after collections rather than individuals, we can see how this might be surprising on a first reading of Section 1. We can add a forward reference from Section 1 to the discussion in Section 2.1. When developing our notation, we discussed this point at length, soliciting input from many members of the community. We ultimately decided on collections rather than individuals because there are cases where individual names don't work (e.g., height and width are both measured in pixels). While there are also cases where collective names don't work, it's always easy to pluralize the individual name (e.g., chans). We'll be happy to add this rationale to Section 2.1.
>
> Thank you for your other requested changes. We agree with all of them and will fix them, if they have not been fixed already.

---

### Decision · Action_Editors · 2022-12-11

**Recommendation:** Accept as is

**Comment:**

Reviewers agree that the proposed notation is well-designed and expressive. The reviewers appreciated the author responsiveness in the review process and feel that after recent updates, the paper is correct and clearly relevant to the TMLR community. Improved practices around communicating tensor notation would be valuable for the community, so we hope to see widespread adoption.

**Audience:**

Yes. Tensor notation is used broadly across machine learning.

**Claims And Evidence:**

The original submission had some imprecision, but the reviewers (particularly qUcr) were very thorough, and after the authors have addressed all of the raised issues, we are confident that the paper is clear and correct.